# Synthetic lethality of RB1 and aurora A is driven by stathmin-mediated disruption of microtubule dynamics

Junfang Lyu[1], Eun Ju Yang[1], Baoyuan Zhang[1], Changjie Wu[1], Lakhansing Pardeshi[1], Changxiang Shi[1], Pui Kei Mou[1], Yifan Liu[1], Kaeling Tan[1] & Joong Sup Shim [1✉]

RB1 mutational inactivation is a cancer driver in various types of cancer including lung cancer, making it an important target for therapeutic exploitation. We performed chemical and genetic vulnerability screens in RB1-isogenic lung cancer pair and herein report that aurora kinase A (AURKA) inhibition is synthetic lethal in RB1-deficient lung cancer. Mechanistically, $RB1^{-/-}$ cells show unbalanced microtubule dynamics through E2F-mediated upregulation of the microtubule destabilizer stathmin and are hypersensitive to agents targeting microtubule stability. Inhibition of AURKA activity activates stathmin function via reduced phosphorylation and facilitates microtubule destabilization in $RB1^{-/-}$ cells, heavily impacting the bipolar spindle formation and inducing mitotic cell death selectively in $RB1^{-/-}$ cells. This study shows that stathmin-mediated disruption of microtubule dynamics is critical to induce synthetic lethality in RB1-deficient cancer and suggests that upstream factors regulating microtubule dynamics, such as AURKA, can be potential therapeutic targets in RB1-deficient cancer.

[1] Cancer Centre, Faculty of Health Sciences, University of Macau, Avenida da Universidade, Taipa, Macau, SAR, China. ✉email: jsshim@um.edu.mo

Retinoblastoma susceptibility gene (RB1) is the first tumor suppressor identified in retinoblastoma, a rare form of intraocular cancer in children[1]. RB1 has emerged as a critical regulator of various biological processes[2]. The canonical role of RB1 is to regulate cell cycle progression via suppressing the transcription of E2F target genes[3,4]. RB1 directly binds to the transcription activation domain of E2F transcription factor family and inhibits E2F-driven transcription of target genes. It also recruits corepressor complex to the promoter of E2F target genes and represses E2F-driven target gene transcription. Upon mitogen signaling, activated cyclin/CDK complexes sequentially phosphorylate RB1 and dissociate RB1 from E2F, activating E2F target genes and prompting cells enter G1 cell cycle. RB1 remains inactive during the rest of cell cycle until cells enter mitosis and protein phosphatases dephosphorylate RB1[5–7]. Due to its crucial role in cell cycle control, mutational inactivation of RB1 is a cancer driver in many types of cancer. RB1 mutations are found in a variety of cancer at variable frequencies. Lung cancer is one of the top carriers of mutant RB1, with 90% of mutation frequency found in small cell lung cancer (SCLC) and 20% frequency found in non-small cell lung cancer (NSCLC)[8]. Unlike NSCLC, no targetable oncogene has been identified in SCLC. Therefore, mutant RB1 became an important target for therapeutic exploitation for SCLC, despite its nature of tumor suppressor mutations.

Aurora kinase A (AURKA) is a serine/threonine kinase whose activity peaks during G2/M transition in the cell cycle[9]. It interacts with and phosphorylates a number of target proteins during the mitotic spindle assembly, centrosome separation, and cytokinesis, hence playing a critical role in mitosis[10]. A number of recent studies demonstrated that AURKA is frequently overexpressed in a wide spectrum of cancers[11–16], and its overexpression is associated with poor clinical outcomes in cancer patients[17,18], rendering it a highly important cancer target. It has also been identified as a synthetic lethal target for several tumor suppressors, including ARID1A[19], SNF5[20], and SMARCA4[21], as well as RB1 very recently[22].

Synthetic lethality or induced essentiality is one of very few approaches that enable to target tumor suppressor mutations with pharmacological agents[23]. In order to identify RB1 synthetic lethal targets, we conducted chemical and genetic vulnerability screenings using epigenetics-focused small molecules and RNAi libraries in RB1-isogenic lung cancer cells. We here report that AURKA is amongst the strongest synthetic lethal candidate for RB1 deficiency in lung cancer cells. Our mechanistic exploration of the synthetic lethality between RB1 and AURKA reveals that the E2F target stathmin is highly upregulated in RB1-deficient cells where it induces unbalanced microtubule dynamics. Stathmin is a microtubule destabilizer and a substrate of AURKA for inhibitory phosphorylation. Inhibition of AURKA activity activates stathmin and strongly facilitates microtubule depolymerization in RB1-deficient cancer cells, severely impacting the spindle formation and triggering mitotic cell death. Based on our findings, we suggest that upstream factors regulating microtubule dynamics are promising drug targets in lung cancer cells with RB1 loss-of-function mutations.

## Results

### $RB1^{-/-}$ lung cancer cells are vulnerable to AURKA inhibition.
In order to screen for RB1 synthetic lethal targets, we first generated RB1 knockout ($RB1^{-/-}$) cell lines from the RB1 wildtype A549 and HCC827 lung cancer cells using a CRISPR/Cas9 system (Supplementary Fig. 1a–c). RB1 knockout was verified with Sanger sequencing of genomic RB1 locus and RB1 Western blots and immunofluorescence (Fig. 1a, b; Supplementary Fig. 1d–h).

The functional knockout of RB1 in $RB1^{-/-}$ cells was verified with canonical RB1-E2F targets, CDK2, and cyclin E expression[24,25] (Supplementary Fig. 1e). There was no significant difference in cell proliferation rate between $RB1^{+/+}$ and $RB1^{-/-}$ cell pairs (Supplementary Fig. 2a, b). To identify synthetic lethality with RB1 loss in lung cancer cells, we selected libraries of epigenetics RNAi (siRNA library targeting 463 human epigenetics machineries with a pool of 4 siRNAs for each target) and epigenetics compounds (128 small molecule inhibitors of various epigenetics machineries) due to the functional relationship between RB1/E2F axis and epigenetics machineries in transcription regulation. The epigenetics RNAi screening was done in 50 nM to ensure gene silencing of the wide variety of siRNA targets. The GAPDH siRNA was included across the plates for the quality control of the gene silencing efficiency during the screening. The epigenetics small molecule screening was done with an 8-dose inter-plate titration format (14 nM – 30 μM) in 384-well plates to cover wide dosage range and get accurate $IC_{50}$ values (Fig. 1c). In the RNAi screening, we found 3 candidate synthetic lethal genes that have a Z score of less than −3, including AURKA, HDAC1, and MYC (Fig. 1d, e). In the small molecule screening, we found 11 candidates (5 classes of inhibitors) that have a selectivity index (SI) bigger than 4, including 5 AURKA inhibitors (such as ENMD-2076, VX-689, Alisertib, AMG-900, Tozasertib), 2 BET inhibitors, 2 HDAC inhibitors, a JAK2 inhibitor, and a HIF inhibitor (Fig. 1f, g). AURKA was the top synthetic lethal candidate that commonly appeared from the both screenings. AURKA is known to phosphorylate well-known epigenetic regulators, heterochromatin protein 1γ (HP1γ) at Ser83 and histone H3 at Thr 118, to regulate chromatin structure and gene expression networks[26,27], thus being included in the epigenetics libraries. Among the AURKA inhibitors, we mainly used ENMD-2067 in follow-up studies as it appeared to be the best synthetic lethal hit from the screen. We also used other selective AURKA inhibitors, such as alisertib and Aurora A Inhibitor I (TC-S 7010), as well as an AURKA specific siRNA, to cross validate the ENMD-2076 effects. We then tested the synthetic lethality between RB1 and AURKA with various concentrations of AURKA siRNA and small molecule AURKA inhibitors on A549 and HCC827 RB1-isogenic cell pairs, verifying the screening results (Fig. 1h–j; Supplementary Fig. 2c–f). We next tested AURKA inhibition in a panel of lung cancer cell lines with different RB1 status and found that the synthetic lethal effect appeared in general in RB1-mutant, SCLC cell lines (Fig. 1k–m; Supplementary Fig. 2g). To exclude the possibility that the synthetic lethal phenotype induced by AURKA inhibitors was a general mitotic kinase inhibitory effect in RB1-deficient cells, we tested inhibitors of other mitotic proteins, such as TTK/Mps1, PLK1, and Eg5, in the RB1-isogenic pair. Unlike AURKA inhibitors, these mitotic inhibitors did not show significant synthetic lethal effect in RB1-deficient lung cancer cells, suggesting that the synthetic lethality by AURKA inhibitors was not due to the general mitotic kinase inhibitory effect (Supplementary Fig. 3a–c).

In order to test the synthetic lethal effect in vivo, A549 and HCC827 isogenic lung cancer xenografts were established in nude mice. A low dose ENMD-2076 (25 mg/kg) treatment did not inhibit the growth of $RB1^{+/+}$ A549 tumor xenografts, while a high dose (50 mg/kg) marginally inhibited it (Fig. 2a). However, both dosages of ENMD-2076 almost completely inhibited the growth of $RB1^{-/-}$ A549 tumor xenografts (Fig. 2b, c). Similar effect was observed in HCC827 tumor xenograft experiments where ENMD-2076 selectively inhibited the growth of $RB1^{-/-}$ tumors (Fig. 2d–f). Alisertib and Aurora A Inhibitor I also showed selective antitumor effects on $RB1^{-/-}$ lung cancer xenografts (Fig. 2g–i; Supplementary Fig. 4a–i). From the analyses of tumor samples, we observed that AURKA inhibitor treatment

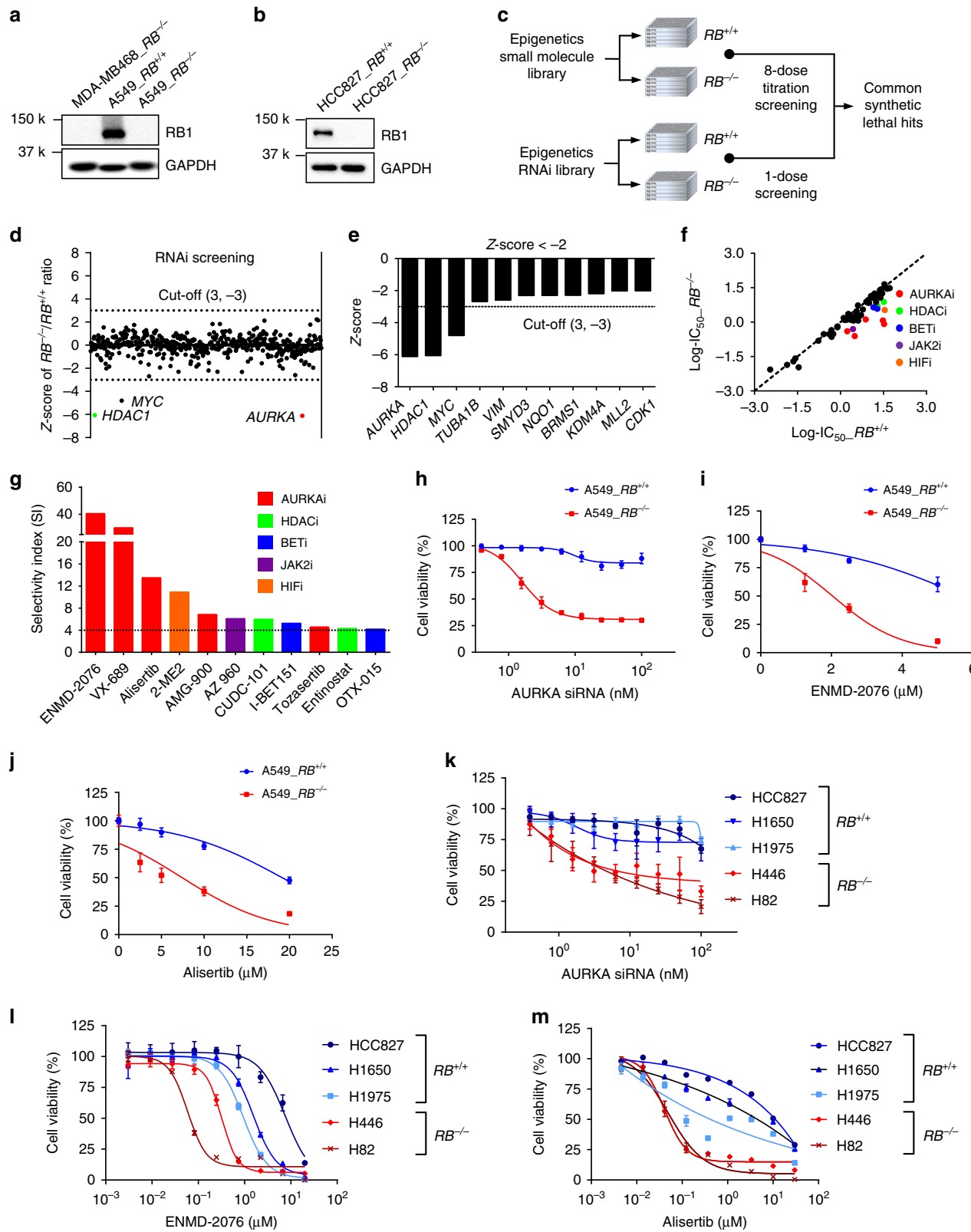

selectively induced caspase-3 activation and inhibited tumor cell proliferation in $RB1^{-/-}$ lung cancer xenografts in mice without apparent body weight changes (Fig. 2j, k; Supplementary Fig. 5a–h; Supplementary Fig. 6a–d), indicating that RB1 loss greatly increased the vulnerability of the cancer cells to AURKA inhibition in vivo.

**$RB1^{-/-}$ cells have unbalanced microtubule dynamics**. While we were generating RB1-KO lung cancer clones, we have constantly observed that α-tubulin protein level was reduced when RB1 expression was absent or reduced in lung cancer cells (Supplementary Fig. 7a). We also observed the same result in mice tumor tissues where α-tubulin level was overall reduced in $RB1^{-/-}$

**Fig. 1 Identification of AURKA as a synthetic lethal partner of RB1 in lung cancer cells. a, b** Western blot analyses to verify RB1 knockout in $RB1^{-/-}$ lung cancer cells. MDA-MB468, a RB1-null breast cancer cell line, was used as a positive control for RB1 knockout. GAPDH was used as a loading control. **c** Flow chart of the synthetic lethality screenings in epigenetics small molecule and RNAi libraries. **d** Scatter plot of the averaged $Z$-scores for epigenetic siRNA screening. The $Z$-score of $-3$ was used as the cut-off (dotted line) to identify synthetic lethality hits. The three top hits, AURKA, HDAC1, and MYC are shown with red, green, and black dots, respectively. The screening was done in duplicate. **e** Total 11 candidates were found as potential synthetic lethality partners with the averaged $Z$-scores less than $-2$. The cut-off of $Z$-score $-3$ is marked with a dotted line. **f** Scatter plot of averaged log-$IC_{50}$ values of each small molecule for A549 $RB1^{+/+}$ and $RB1^{-/-}$ from small molecule screening. The diagonal line represents small molecules that do not exhibit any selectivity toward A549 $RB1^{+/+}$ or $RB1^{-/-}$ cell lines. Colored dots are candidate small molecules that selectively inhibited the viability of A549 $RB1^{-/-}$ cells. The average $IC_{50}$ values from the duplicate screenings are shown. **g** Selectivity indices (SI) of the synthetic lethal candidates for RB1. $SI = IC_{50}^{\ RB1\ +/+}/IC_{50}^{\ RB1-/-}$. Top 11 candidates with SI >4 (dotted line) are shown in the graph. Validation of the synthetic lethality between RB1 and AURKA using AURKA siRNA (**h**) and small molecule AURKA inhibitors, including ENMD-2076 (**i**) and alisertib (**j**). Data are presented as mean ± SD ($n = 3$ independent experiments). Synthetic lethal effects of AURKA siRNA (**k**), ENMD-2076 (**l**) and alisertib (**m**) in a panel of lung cancer cell lines with different RB1 status. The cell viability was measured with AlamarBlue assay. Data are presented as mean ± SD ($n = 3$ independent experiments).

tumors, and was further reduced by the AURKA inhibitor treatment (Fig. 2k; Supplementary Fig. 5d, g, h) or AURKA silencing (Supplementary Fig. 2c). Since the differential expression level of α-tubulin was observed in RB1-isogenic cell lines, we hypothesized that RB1 loss caused the reduction of the α-tubulin level. We then analyzed effects of RB1 loss and AURKA inhibition on α-tubulin level in greater detail. Indeed, α-tubulin protein level was much lower in $RB1^{-/-}$ than in $RB1^{+/+}$ A549 cells, and AURKA inhibitors (ENMD-2076 and Aurora A Inhibitor I) or siRNA further reduced the α-tubulin level (Fig. 3a, b; Supplementary Fig. 7c). Same results were observed in HCC827 RB1 isogenic cell pair (Supplementary Fig. 7b). Alisertib, in contrast, slightly increased the α-tubulin level (Supplementary Fig. 5e, f; Supplementary Fig. 7d). To test whether RB1 dynamically regulates the α-tubulin level, we transiently silenced RB1 in $RB1^{+/+}$ cells and observed the α-tubulin level. The α-tubulin level was significantly reduced by RB1 silencing and was further reduced by the co-silencing of AURKA (Fig. 3c). The α-tubulin protein level is largely regulated either by autoregulatory control of α-tubulin mRNA stability or proteasome-mediated degradation of the protein[28,29]. To test whether the reduction of the α-tubulin level occurred at the post-transcriptional level or at the post-translational level, we analyzed the RT-qPCR of the α-tubulin transcript and the half-life of the α-tubulin protein. RB1 loss did not heavily affect the level of α-tubulin mRNA, while it significantly reduced the half-life of the α-tubulin protein (Fig. 3d–f), suggesting that RB1 loss reduced the α-tubulin protein stability.

α-tubulin is a subunit that forms a dimer with β-tubulin to form the macromolecular polymerized structure, microtubule. The α-tubulin protein is generally more stable in polymerized state than in depolymerized state because tubulin dimers from depolymerized microtubule have higher chance to be subjected to the proteasome-dependent degradation[29–31]. In order to examine whether the reduced α-tubulin stability in $RB1^{-/-}$ cells was due to the reduced microtubule polymerization in the cells, we analyzed the polymerization status of microtubules in the RB1-isogenic lung cancer cell pairs. The result showed that $RB1^{-/-}$ cells have significantly less polymerized microtubules than in $RB1^{+/+}$ cells (Fig. 3g–i). These results suggested that RB1 loss caused an unbalance in the microtubule dynamics, leading to the microtubule depolymerization and hence reduced α-tubulin protein level. Given that AURKA inhibition in $RB1^{-/-}$ cells further reduced α-tubulin level resulting in the significant reduction of the α-tubulin level in the cells, the microtubule dynamics could be a possible mediator of the synthetic lethality between RB1 and AURKA. To test this hypothesis, we treated the RB1-isogenic cell pair with α-tubulin siRNA or small molecules targeting microtubule stability, including vinorelbine and paclitaxel, and observed

synthetic lethal effect. The silencing of α-tubulin expression did not induce synthetic lethality, while vinorelbine, a microtubule destabilizer, strongly induced synthetic lethality in $RB1^{-/-}$ cells (Fig. 3j–m). Paclitaxel, a microtubule stabilizing agent, also induced synthetic lethality in $RB1^{-/-}$ cells, albeit to a lesser extent (Fig. 3l, n). Vinorelbine reduced α-tubulin level, while paclitaxel increased it, further indicating that the α-tubulin protein stability was regulated by the change in the microtubule dynamics. These results indicated that RB1-deficient cells have unbalanced microtubule dynamics and are hyper-vulnerable to the agents that change the microtubule stability.

**Stathmin is up-regulated in $RB1^{-/-}$ lung cancer cells.** To identify factor(s) that leads to the unbalanced microtubule dynamics in RB1-deficient cells, we analyzed transcriptome profiles in A549 RB1-isogenic cell pair. Gene ontology (GO) analysis of the differentially expressed genes revealed that 10 tubulin-binding proteins, including *STMN1*, *STMN3*, *EML2*, *TPPP*, *TPPP3*, *NCALD*, *CLIP1*, *IFT81*, *PARK2*, and *LRRK2*, were significantly up- or down-regulated in $RB1^{-/-}$ cells (Fig. 4a, b). To exclude genes that were differentially expressed in a cell context-dependent manner, we analyzed the 10 genes in E2F ChIP-Seq database (ChIP-Atlas, http://chip-atlas.org/) for the prediction of E2F target genes. ChIP-Atlas predicts genes directly regulated by given proteins, based on binding profiles of all public ChIP-seq data for particular gene loci[32]. Among the candidates, *STMN1*, which encodes stathmin (also known as Op18), showed the top binding score for E2F1 on its promoter (Fig. 4c). Chen et al. have also reported previously that E2F1 could transactivate stathmin in hepatocellular carcinoma cells[33]. Stathmin is known to bind preferentially to unpolymerized tubulin dimers and increase the microtubule catastrophe rate[34]. To further validate whether stathmin is the target of E2F1, we conducted a series of experiments showing the expression regulation of stathmin by RB1 loss. We observed the overexpression of stathmin in $RB1^{-/-}$ cells (Fig. 4d, e), the overexpression of stathmin by RB1 silencing in $RB1^{+/+}$ cells (Fig. 4f), and the down-regulation of stathmin by E2F1 silencing in $RB1^{-/-}$ cells (Fig. 4g). The α-tubulin level was negatively correlated with the stathmin level, suggesting the role of stathmin in microtubule destabilization (Fig. 4d–f). We also verified the E2F1 binding to the stathmin promoter with the primer pair corresponding to the predicted E2F1 binding site on the proximal promoter (Fig. 4h). In addition, E2F2 and 3 could also bind to the proximal promoter of stathmin (Fig. 4i, j), suggesting that stathmin is a transcription target of E2F family transcription factors and its expression is negatively regulated by RB1. Next, to test whether the negative regulation of stathmin by RB1 occurs in general, not in a cell context-dependent, we analyzed the protein levels of RB1 and stathmin in a panel of lung

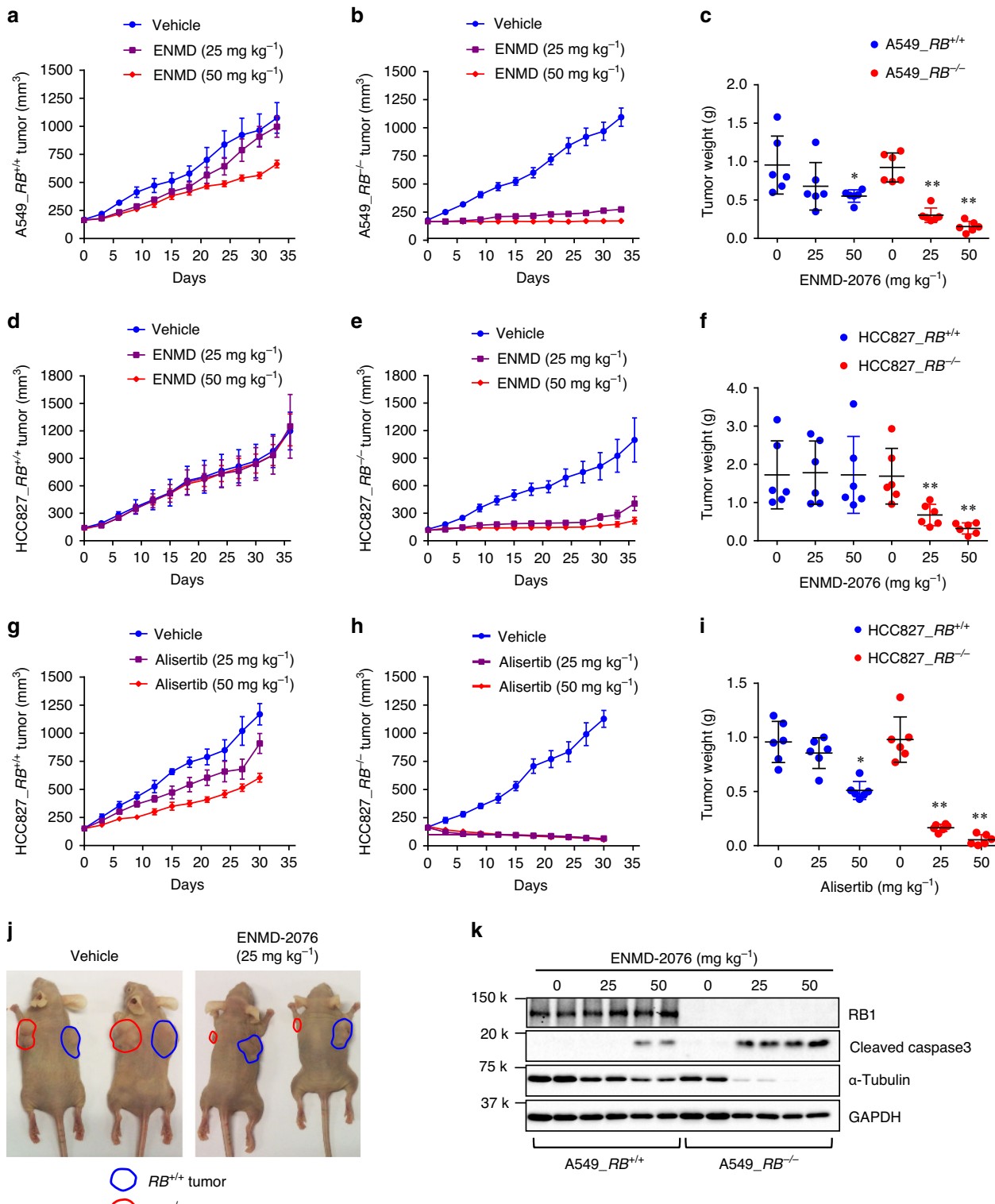

**Fig. 2 AURKA inhibitors induce synthetic lethality in $RB1^{-/-}$ lung cancer in vivo.** Tumor growth curve of the A549 $RB1^{+/+}$ (**a**) and $RB1^{-/-}$ (**b**) lung cancer xenografts in mice treated with ENMD-2076. **c** Endpoint weight measurement of tumors isolated from the nude mice bearing A549 $RB1^{+/+}$ and $RB1^{-/-}$ lung cancer xenografts. Tumor growth curve of the HCC827 $RB1^{+/+}$ (**d**) and $RB1^{-/-}$ (**e**) lung cancer xenografts in mice treated with ENMD-2076. **f** Endpoint weight measurement of tumors isolated from nude mice bearing HCC827 $RB1^{+/+}$ and $RB1^{-/-}$ lung cancer xenografts. Tumor growth curve of the HCC827 $RB1^{+/+}$ (**g**) and $RB1^{-/-}$ (**h**) lung cancer xenografts in mice treated with alisertib. **i** Endpoint weight measurement of tumors isolated from nude mice bearing HCC827 $RB1^{+/+}$ and $RB1^{-/-}$ lung cancer xenografts. Data are presented as mean ± SD ($n = 6$ independent animals per group). *$P < 0.05$; **$P < 0.01$ vs no treatment control, determined using two-sided Student's $t$ test. **j** Representative images of vehicle-treated and ENMD-2076-treated nude mice bearing A549 $RB1^{+/+}$ and $RB1^{-/-}$ lung cancer xenografts. Tumor areas are indicated with dotted lines. **k** Western blot analyses of expression levels of RB1, cleaved caspase3, and α-tubulin in the tumor tissues. GAPDH was used as a loading control.

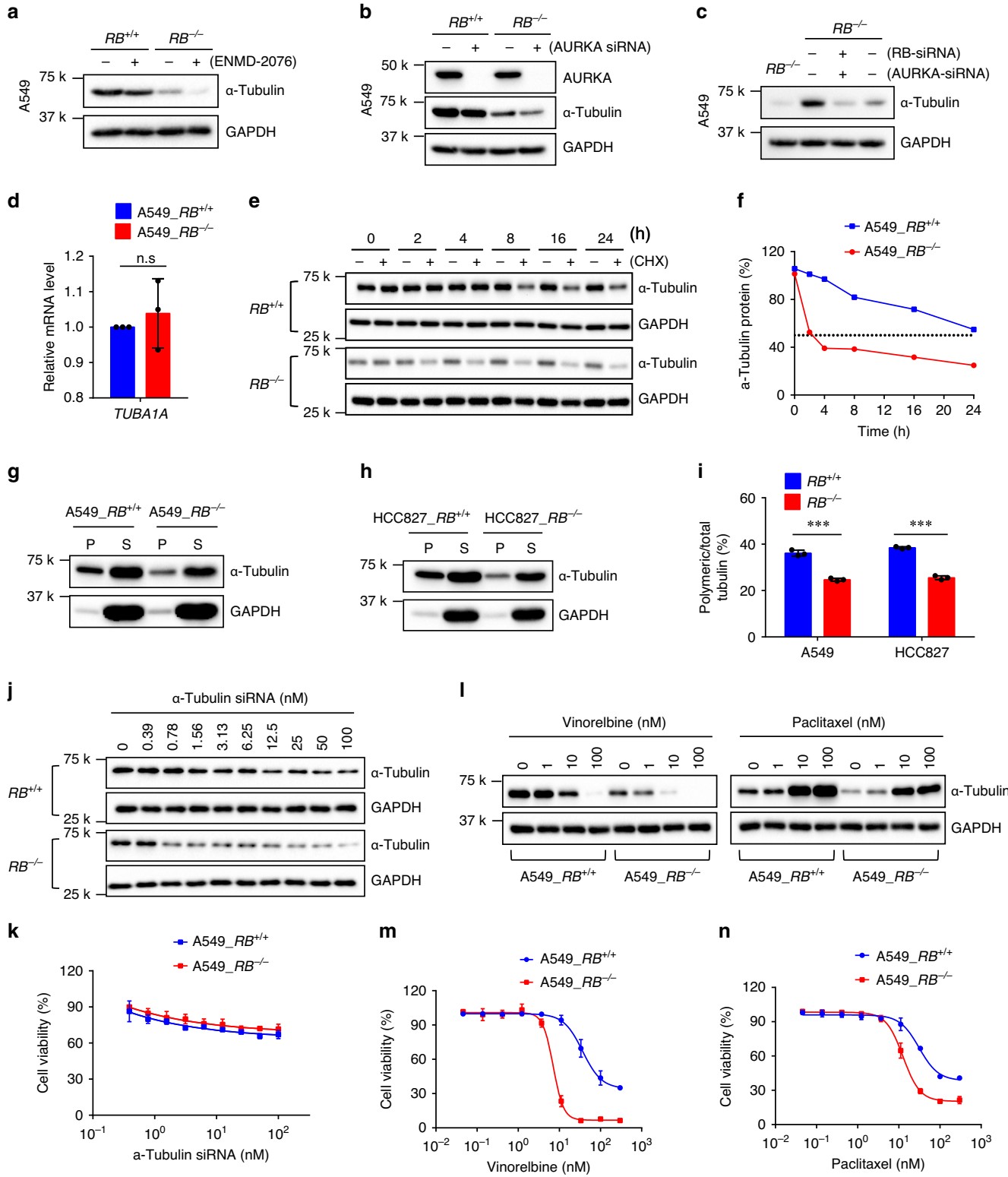

cancer cell lines, including wildtype *RB1* expressing cells (A549, HCC827, H1975, and H1650) and *RB1*-null cells (H446 and H82). Among the 6 lung cancer lines tested, the two RB1-null SCLC cell lines, H446 and H82, showed the highest level of stathmin (Fig. 4k). We also analyzed 365 lung adenocarcinoma patients' samples from TCGA database for the protein expression status, and found that stathmin was ranked at the top 5 among

the proteins whose expression was negatively correlated with RB1 protein level (Fig. 4l, m), further verifying our observations.

**AURKA inhibition facilitates microtubule destabilization.** Stathmin is a phosphoprotein in which the phosphorylation results in the loss of the microtubule-destabilizing activity[35–37]. Two serine phosphorylation sites, Ser16 and Ser63, in stathmin

**Fig. 3 Disruption of the microtubule dynamics causes synthetic lethality in $RB1^{-/-}$ lung cancer cells. a, b** Reduction of the α-tubulin level in $RB1^{-/-}$ cells and its further reduction by AURKA inhibition. A549 $RB1^{+/+}$ and $RB1^{-/-}$ cells were treated with 2.5 μM ENMD-2076 (**a**) for 24 h or 25 nM AURKA siRNA (**b**) for 48 h. **c** A549 $RB1^{+/+}$ cells were transfected with RB siRNA with or without AURKA siRNA for 48 h and analyzed for Western blotting of α-tubulin. **d** RT-qPCR analysis of TUBA1A and TUBA4A mRNA levels in A549 $RB1^{+/+}$ and $RB1^{-/-}$ cells. Data are presented as mean ± SEM ($n = 3$ independent experiments). n.s denotes not significant, determined using one sample $t$ test with the hypothetical value = 1. **e, f** Measurement of α-tubulin half-life in A549 $RB1^{+/+}$ and $RB1^{-/-}$ cells. Cells were treated with or without 10 μM cycloheximide (CHX) for the indicated time points and analyzed for α-tubulin protein level (**e**). The α-tubulin level was quantified and normalized to the GAPDH level from two Western blot data (**f**). The dotted line indicates the half-life of the α-tubulin protein. **g-i** Analysis of polymeric microtubules in $RB1^{+/+}$ and $RB1^{-/-}$ lung cancer cells. A549 (**g**) or HCC827 (**h**) RB1-isogenic cell pairs were lysed and separated into the soluble (S) and polymerized (P) microtubule fractions. The samples were subjected to Western blot with α-tubulin antibody. GAPDH was used as a loading control. The α-tubulin protein level was quantified and the ratio of the polymerized/total α-tubulin was calculated (**i**). Data are presented as mean ± SEM ($n = 3$ independent experiments). ***$P < 0.001$ between two indicated groups, determined using two-sided Student's $t$ test. The effect of α-tubulin siRNA, vinorelbine (microtubule destabilizer), and paclitaxel (microtubule stabilizer) on the viability of A549 $RB1^{+/+}$ and $RB1^{-/-}$ lung cancer cells. A549 $RB1^{+/+}$ and $RB1^{-/-}$ cells were treated with the indicated concentrations of α-tubulin siRNA (**j, k**), vinorelbine (**l, m**), or paclitaxel (**l, n**) for 72 h, and the α-tubulin protein level (**j, l**) and cell viability (**k, m**, and **n**) were analyzed. Data are presented as mean ± SD ($n = 3$ independent experiments).

contain a consensus sequence for AURKA phosphorylation and the mutations in these two serine sites abolished stathmin phosphorylation by AURKA, suggesting that stathmin is a substrate of AURKA for phosphorylation[38,39]. In order to explore a possible crosstalk between AURKA and stathmin functions in microtubule dynamics, we tested the effect of AURKA inhibition on stathmin phosphorylation. Stathmin phosphorylation was only detectable in nocodazole-synchronized cells, suggesting its high phosphorylation during mitosis (Fig. 5a). AURKA activity also peaks during G2/M phase[40]. Inhibition of AURKA kinase activity by ENMD-2076, as well as by alisertib or Aurora A Inhibitor I, significantly reduced the stathmin phosphorylation at Ser16 (Fig. 5a; Supplementary Fig. 8a, b). AURKA silencing also completely shut down the stathmin phosphorylation at Ser16 (Fig. 5b), suggesting that AURKA activity is required for the phosphorylation of stathmin at Ser16 during mitosis. Since stathmin dephosphorylation is known to activate its activity to promote microtubule destabilization, we next tested the effect of AURKA inhibition on microtubule polymerization. Vinorelbine and paclitaxel were used as positive controls for microtubule polymerization status. Treatment of A549 cells with ENMD-2076 or Aurora A inhibitor I reduced the polymerized microtubules at the similar extent to that seen by vinorelbine, while paclitaxel treatment highly increased the polymerized microtubules (Fig. 5c–e; Supplementary Fig. 9a, b). Similar effects were observed in HCC827 cells treated with ENMD-2076 (Supplementary Fig. 9c, d). Notably, like all other AURKA inhibitors, alisertib also significantly reduced the microtubule polymerization (Supplementary Fig. 9e, f), even though it increased the α-tubulin protein level (Supplementary Fig. 5e, f; Supplementary Fig. 7d). The increase in α-tubulin level by alisertib was presumably due to its unique off-target effect. From a recent DNA-programmed affinity labeling method, alisertib was found to directly bind to an α-tubulin isoform[41], supporting this notion. AURKA silencing also reduced polymerized microtubules in A549 RB1-isogenic cells lines (Fig. 5f, g). We also confirmed that stathmin silencing in $RB1^{-/-}$ cells could promote microtubule polymerization (Fig. 5h–j) and its overexpression in $RB1^{+/+}$ cells facilitated microtubule depolymerization (Fig. 5k–m). These data suggested that the inhibition of AURKA activity reduced the stathmin phosphorylation and in turn promoted the stathmin function in microtubule depolymerization, thus disrupting microtubule dynamics, especially in $RB1^{-/-}$ cells where stathmin was overexpressed and microtubule dynamics was already unbalanced. To further visualize the microtubule polymerization status in cells, we adopted a method developed by Harkcom and colleagues that used anti-Tyr-tubulin immunostaining to measure

the microtubule polymerization index[42]. Our result showed that AURKA inhibition, as well as siRNA silencing, significantly reduced microtubule polymerization index, a phenotype similar to that seen in vinorelbine treatment (Fig. 6a–d).

Given the critical roles of AURKA and stathmin in mitosis, the disruption of microtubule dynamics by the inhibition of AURKA-stathmin pathway would be detrimental to the spindle assembly in mitotic $RB1^{-/-}$ cells. We thus examined mitotic spindle formation in RB1-isogenic cells treated with the AURKA inhibitor or microtubule targeting agents. In $RB1^{+/+}$ cells, ENMD-2076 did not heavily affect spindle bipolarity but shortened the spindle length (small-bipolar) (Fig. 7a, b). Vinorelbine induced a similar small-bipolar spindle morphology with increased number of monopolar spindle cells in $RB1^{+/+}$ cells. Paclitaxel, however, induced multipolar spindle formation in the majority of mitotic cells in both $RB1^{+/+}$ and $RB1^{-/-}$ cells (Fig. 7a–c). In $RB1^{-/-}$ cells, ENMD-2076 and vinorelbine induced monopolar spindles with very short spindle lengths, suggesting that $RB1^{-/-}$ cells lost spindle bipolarity and the microtubule length was shorten. Very similar phenotypes were observed in the RB1-isogenic cell pair treated with either AURKA siRNA (Supplementary Fig. 10a–d) or other AURKA inhibitors, including alisertib and Aurora A Inhibitor I (Supplementary Fig. 11a–c). These data suggested that either AURKA inhibitors or microtubule destabilizers severely affected the spindle assembly and bipolarity in $RB1^{-/-}$ cells, while $RB1^{+/+}$ cells were affected moderately by the inhibitors. In line with these observations, ENMD-2076, as well as other AURKA inhibitors, induced G2/M arrest in $RB1^{+/+}$ cells without mitotic cell death (Fig. 7d, e; Supplementary Fig. 12a, b; Supplementary Fig. 13a, b). Whereas, it heavily induced cell death in $RB1^{-/-}$ cells via inducing apoptosis (Fig. 7f–i; Supplementary Fig. 12c–f; Supplementary Fig. 13c–f). AURKA and PLK1 are involved in G2/M transition and early mitotic entry by activating CDC25C and CDK1[19]. Thus, inhibiting AURKA activity in general induces G2/M arrest[43], similar to the phenotype we observed in $RB1^{+/+}$ cells. However, in $RB1^{-/-}$ cells where spindle assembly checkpoint (SAC) was already primed through RB1 loss-induced MAD2 upregulation[22,44] or through the unbalanced microtubule dynamics by stathmin overexpression, AURKA inhibition could hyperactivate SAC via stathmin-mediated disruption of the microtubule dynamics.

**Synthetic lethality of RB1 and AURKA is mediated by stathmin.** To test whether the synthetic lethality by AURKA inhibition was mediated by stathmin, stathmin was overexpressed or silenced in RB1-isogenic cells and tested AURKA inhibitor sensitivities. Stathmin overexpression significantly sensitized the viability of $RB1^{+/+}$ cells to AURKA silencing (IC$_{50}$ shifted from >100 to

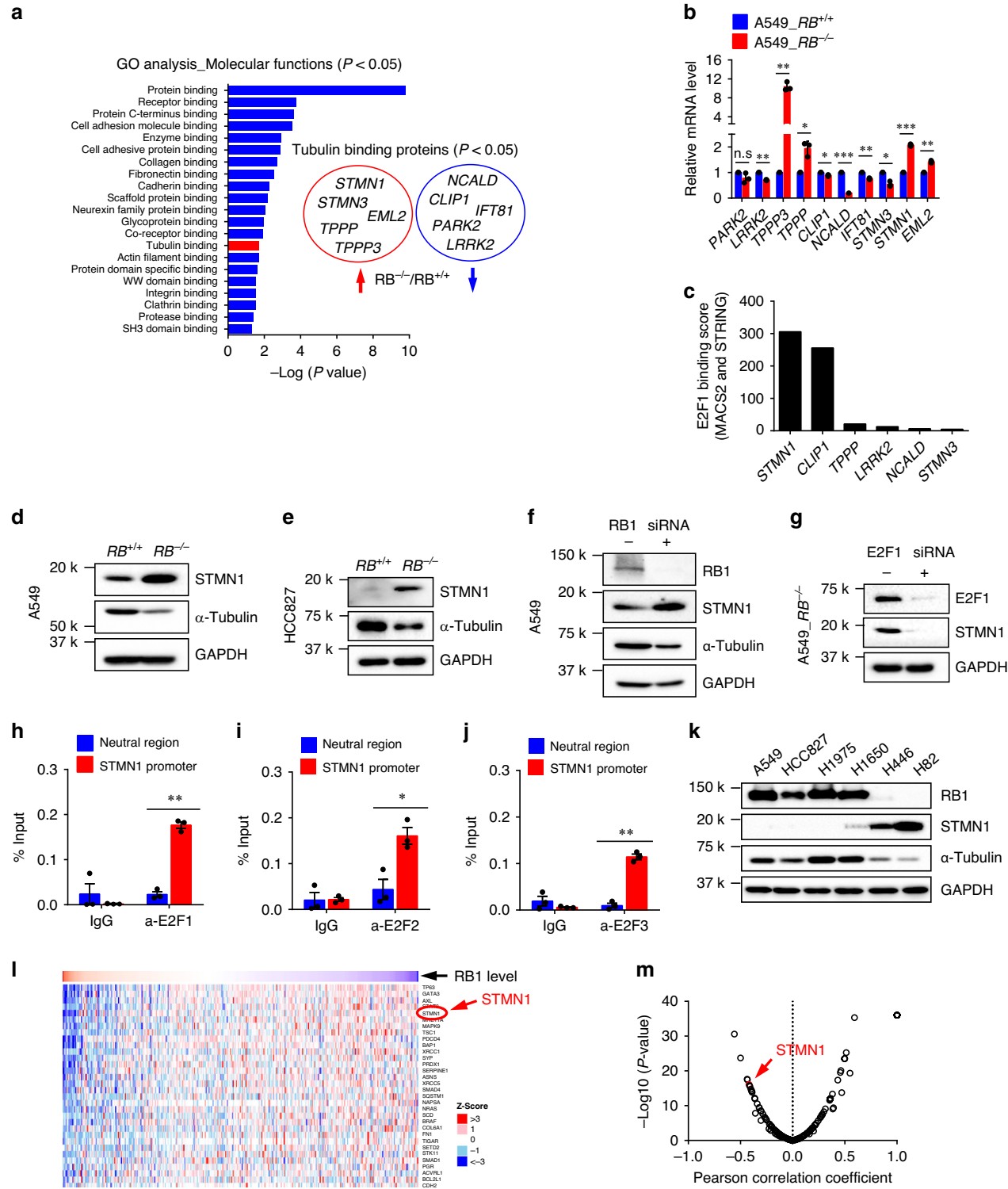

13.8 nM, Fig. 8a). On the other hand, stathmin silencing itself showed inhibitory effect on the cell viability with a marginal selectivity to $RB1^{-/-}$ cells (Fig. 8b). This phenomenon was reminiscent to the effect shown by paclitaxel, a microtubule stabilizer, which also induced a marginal synthetic lethal effect in $RB1^{-/-}$ cells (Fig. 3n). We therefore selected nonlethal concentrations of stathmin siRNA (0.1, 0.5, and 1 nM) to test the rescue effect on AURKA inhibitor-induced synthetic lethality. While the stathmin siRNA did not significantly affect the viability of $RB1^{+/+}$ cells treated with AURKA siRNA (Fig. 8c), it dose-

dependently rescued the synthetic lethal effect of the AURKA siRNA in $RB1^{-/-}$ cells ($IC_{50}$ shifted from 4.9 to 40.2 nM at 1 nM stathmin siRNA, Fig. 8d), suggesting that the synthetic lethal effect of AURKA inhibitors in $RB1^{-/-}$ cells was mediated in part by stathmin.

This study provides strong evidences to support the model that AURKA inhibition promotes the stathmin activity, resulting in the disruption of microtubule dynamics, which sensitizes RB1-deficient lung cancer cells to mitotic cell death. We then hypothesized that these sequential events − AURKA-stathmin-

**Fig. 4 RB1-deficiency activates the E2F-dependent transcription of stathmin. a** Transcriptome profiling of A549 $RB1^{+/+}$ and $RB1^{-/-}$ cells and the gene ontology (GO) analysis by molecular functions for differentially expressed genes between the isogenic cells. Fisher Exact test was performed to categorize the genes in GO groups ($P < 0.05$). Among the differentially expressed genes, tubulin binding proteins (total 10) are shown in the circles as up-regulated (red) or down-regulated (blue) genes in $RB1^{-/-}$ cells. **b** Validation of the expression of tubulin binding proteins with RT-qPCR analysis (mean ± SEM, $n = 3$ independent experiments). *$P < 0.05$; **$P < 0.01$; ***$P < 0.001$ between two indicated groups, determined using one sample $t$ test. n.s denotes not significant. **c** ChIP-Atlas database analysis to predict the E2F1 target genes among the tubulin binding proteins. **d, e** Up-regulation of stathmin (STMN1) level and reciprocal down-regulation of α-tubulin level in $RB1^{-/-}$ lung cancer cells. A549 (**d**) or HCC827 (**e**) $RB1^{+/+}$ and $RB1^{-/-}$ cells were analyzed for stathmin and α-tubulin protein levels. **f** A549 $RB1^{+/+}$ cells were transfected with RB1 siRNA and analyzed for stathmin and α-tubulin protein levels. **g** A549 $RB1^{-/-}$ cells were transfected with E2F1 siRNA and analyzed for stathmin protein level. ChIP analysis of the STMN1 promoter in A549_$RB^{-/-}$ cells using a normal mouse IgG, anti-E2F1 (α-E2F1) (**h**), anti-E2F2 (**i**) or anti-E2F3 (**j**) antibody and the primer sets specific for a predicted E2F1, E2F2 or E2F3 binding site on the promoter of human stathmin (STMN1 promoter) or the 10 kb 5'-upstream of the stathmin transcription start site (Neutral region) as a background. Data are presented as percent chromatin precipitated by IgG and E2F1 antibodies at both locations (mean ± SEM, $n = 3$ independent experiments). *$P < 0.05$; **$P < 0.01$ between two indicated groups, determined using two-sided Student's $t$ test. **k** The protein levels of RB1, stathmin and α-tubulin in a panel of lung cancer cell lines. **l, m** The RB1 and stathmin protein expression levels from 365 lung cancer patients were downloaded from TCGA project archived in LinkedOmics (http://www.linkedomics.org/). The data were shown in the heat map (**l**) and volcano plot (**m**), and stathmin is indicated with the red arrows.

microtubule dynamics in RB1-deficient cells — would eventually trigger SAC, a phenotype that has been observed by Gong and colleagues[22]. To further verify this, we tested the ability of the silencing of the key mitotic checkpoint kinase BUB1B to rescue AURKA inhibition-induced synthetic lethality. The level of BUB1B was slightly elevated in $RB1^{-/-}$ cells (Fig. 8e). As expected, the silencing of BUB1B significantly rescued the cell death induced by AURKA siRNA in $RB1^{-/-}$ A549 cells (Fig. 8f), suggesting that RB1-AURKA synthetic lethality accompanies SAC hyperactivation.

## Discussion
This study identified the hyper-vulnerability of RB1-deficient cells to AURKA inhibition, and mechanistically explored the synthetic lethal crosstalk between RB1 and AURKA pathways in lung cancer cells. In line with our findings, the synthetic lethality between RB1 and AURKA has been first reported very recently by Gong and colleagues[22]. From an elegant mechanistic study, they proposed a model that RB1-deficient cells have primed status of SAC, which requires the high AURKA activity for the cell survival as AURKA is capable of overriding an activated SAC[22]. AURKA inhibition highly sensitizes RB1-deficient cells through mitotic arrest and apoptosis. This model was supported with the rescue effect on the synthetic lethality by the depletion of the mitotic checkpoint genes, including *BUB1B* and *BUB3*. However, it remains obscure what factors mediate the SAC activation in RB1-deficient cells treated with AURKA inhibitors. RB1 family proteins, as well as TP53, play an important role in maintaining chromosomal integrity by suppressing E2F-mediated upregulation of the mitotic checkpoint protein Mad2[44,45]. Homozygous loss of all three Rb family members (p107, p130, and pRb) in mice activates E2F-dependent transcription of *Mad2*. The overexpression of Mad2 activates SAC and facilitates aneuploidy, a form of chromosome instability, and tumor progression[44]. Schvartzman and colleagues elaborated that the depletion of whole RB family proteins, including RB1, p107, and p130 is required for the full activation of SAC, and the loss of RB1 alone only modestly increased Mad2, which may not be sufficient to activate SAC in vivo[44]. Therefore, to get a complete picture, upstream factors that mediate SAC activation in RB1-deficient cells treated with AURKA inhibitors need to be identified.

Our study identified stathmin-microtubule dynamics as the factor mediating the SAC activation in the RB1-AURKA synthetic lethality. We provide a series of evidence to support this model that: (1) In $RB1^{+/+}$ cells, stathmin expression is tightly regulated by the RB1/E2F complex. In $RB1^{-/-}$ cells, activated E2F upregulates stathmin expression. (2) The overexpression of

stathmin in $RB1^{-/-}$ cells facilitates microtubule depolymerization in cells, such that the cells have unbalanced microtubule dynamics. This unbalance is not likely to be critical for the cell viability in $RB1^{-/-}$ cells because AURKA phosphorylates stathmin and partially suppresses its activity on microtubule stability. (3) When AURKA activity is inhibited, AURKA no longer phosphorylates and suppresses stathmin, thereby activating the stathmin proteins that have been overexpressed and heavily disrupting the microtubule dynamics in $RB1^{-/-}$ cells. (4) The disruption of microtubule dynamics triggers SAC hyperactivation in $RB1^{-/-}$ cells, leading to mitotic cell death, which is consistent with the findings by Gong et al. that the synthetic lethality was rescued by the depletion of mitotic checkpoint genes (summarized in Fig. 8g).

Stathmin (also known as oncoprotein 18, or Op18) is a small 17 kDa protein that physically interacts with tubulin dimers and inhibits microtubule polymerization[34]. It has two functionally distinguished domain structures, including N-terminus that promotes microtubule catastrophe and C-terminus that is required for tubulin dimer binding and sequestering[46]. When cells enter mitosis, stathmin is highly phosphorylated and its microtubule destabilizing activity is turned off. It becomes dephosphorylated when cells are ready to exit mitosis[47]. The timing of stathmin phosphorylation is well correlated with the peak timing of the activity of AURKA whose expression level and kinase activity peak at G2 and mitosis phases[40,48]. Recent studies show that stathmin contains at least four serine phosphorylation sites, including Ser16, Ser25, Ser38, and Ser63, and two of these (Ser16 and Ser63) are phosphorylated by AURKA[38,39], suggesting that stathmin activity can be switched off by the active AURKA, facilitating the microtubule polymerization during mitosis[49]. This notion is strongly supported by our result that AURKA inhibition by the kinase inhibitor treatment or siRNA silencing reduced stathmin phosphorylation during mitosis and significantly promoted microtubule depolymerization.

Stathmin expression is known to be regulated by some oncogenic transcription factors, including MYC and E2F family proteins[33,50,51]. TP53 is known to suppress the stathmin transcription[52]. From the transcriptome profiling and E2F-ChIP Seq database analyses, we demonstrated that RB1 negatively regulates stathmin transcription through the suppression of E2F transcription factors. Furthermore, TCGA proteomic database analysis revealed that stathmin was the top 5 ranked protein whose expression is negatively correlated with the RB1 protein level in lung cancer clinical samples. Our lung cancer cell panel analysis also showed that cells with RB1 loss expressed high level of stathmin and were associated with unbalanced microtubule

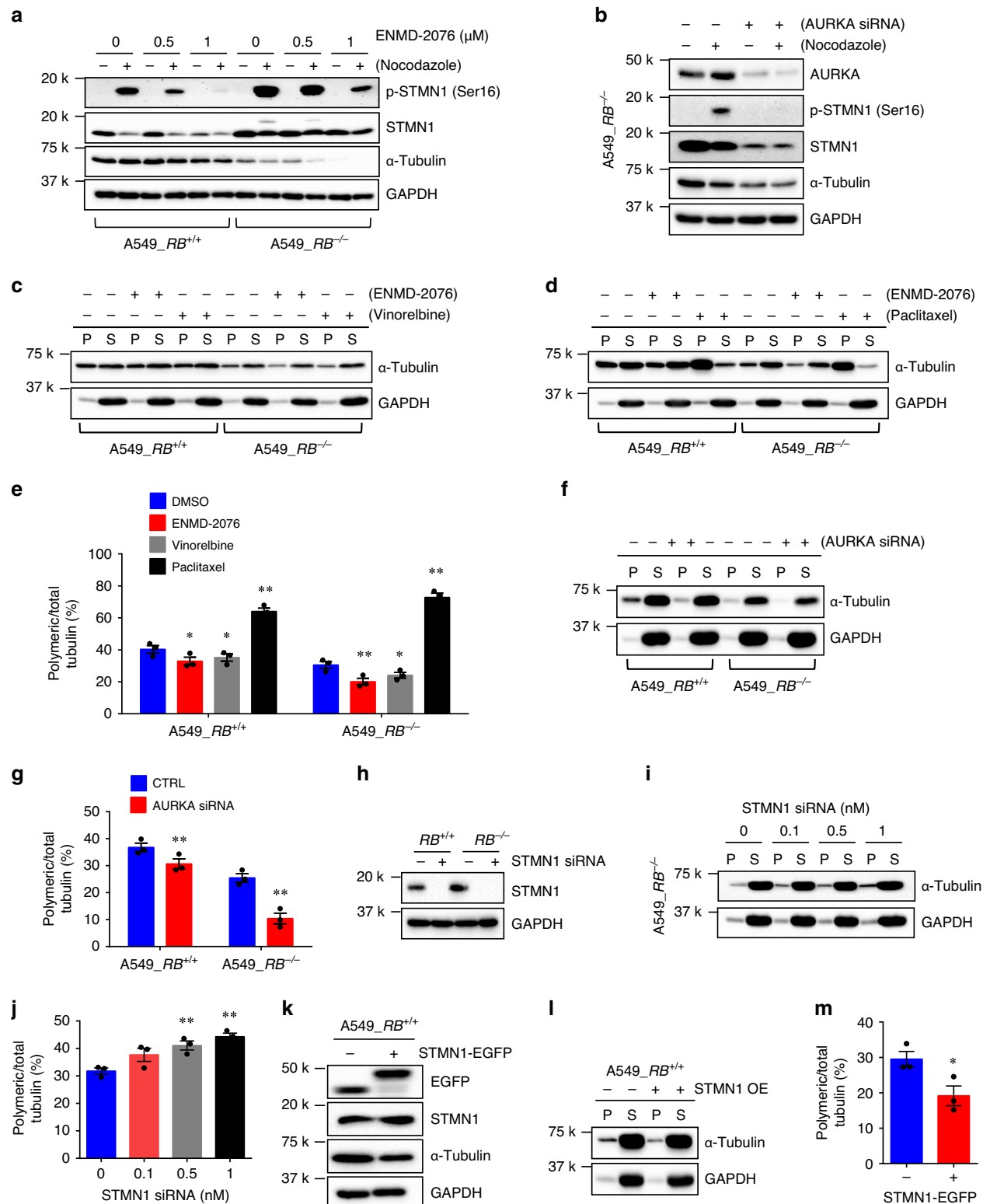

dynamics compared to those expressing wildtype RB1. Based on these results, we propose a synthetic lethality mechanism that RB1-deficient cells highly express a potential weapon (stathmin) that is largely inactivated (phosphorylated form) by the presence of oncogenic AURKA. AURKA inhibition turns the stathmin to a lethal weapon (active, dephosphorylated form) in RB1-deficient cancer cells, killing the cells by disrupting microtubule dynamics.

In summary, this study proposes the agents targeting microtubule dynamics, including AURKA inhibitors, as potential anticancer drugs for the treatment of RB1-deficient cancer, such as SCLC. Since SCLC has no apparent targetable oncogenes discovered, these findings will be of great value in terms of providing precision targets for such types of hard-to-target cancers.

**Fig. 5 AURKA inhibition promotes the stathmin function in microtubule destabilization. a, b** Effect of ENMD-2076 and AURKA siRNA on stathmin phosphorylation. Cells were treated with ENMD-2076 (**a**) or AURKA siRNA (**b**) for 24 h and then 100 nM nocodazole was treated for additional 24 h, prior to the western blot analyses of phospho-stathmin at Ser16, total stathmin, and α-tubulin. **c–e**, Analysis of polymeric microtubules in A549 $RB1^{+/+}$ and $RB1^{-/-}$ cells treated with 2.5 μM ENMD-2076. The microtubule destabilizer vinorelbine (**c**) and the stabilizer paclitaxel (**d**) were used as controls for polymeric microtubule assays. S and P represent soluble and polymerized fractions, respectively. **e** Quantification of the ratio of the polymerized/total α-tubulin. Data are presented as mean ± SEM ($n = 3$ independent experiments). *$P < 0.05$; **$P < 0.01$ vs DMSO control group in each cell line, determined using two-sided Student's $t$ test. **f, g** Effect of AURKA siRNA on the microtubule polymerization. A549 $RB1^{+/+}$ and $RB1^{-/-}$ cells were treated with 25 nM AURKA siRNA for 48 h, prior to do the polymeric microtubule assay. Data are presented as mean ± SEM ($n = 3$ independent experiments). **$P < 0.01$ vs control siRNA group in each cell line, determined using two-sided Student's $t$ test. **h–j** Effect of stathmin siRNA on the microtubule polymerization. Cells were treated with the indicated concentrations of stathmin siRNA for 24 h prior to do the polymeric microtubule assay. The knock-down efficiency of the siRNA (**h**), microtubule polymerization (**i**), and quantitation of the polymeric microtubules (**j**) are shown. Data are presented as mean ± SEM ($n = 3$ independent experiments). **$P < 0.01$ vs control siRNA group, determined using two-sided Student's $t$ test. **k–m** Effect of stathmin overexpression (OE) on the microtubule polymerization. Cells were transfected with 1 μg stathmin-EGFP (STMN1-EGFP) plasmid for 48 h prior to do the polymeric microtubule assay. The overexpression of stathmin-EGFP (**k**), microtubule polymerization (**l**), and quantitation of the polymeric microtubules (**m**) are shown. Data are presented as mean ± SEM ($n = 3$ independent experiments). *$P < 0.05$ vs no overexpression control group, determined using two-sided Student's $t$ test.

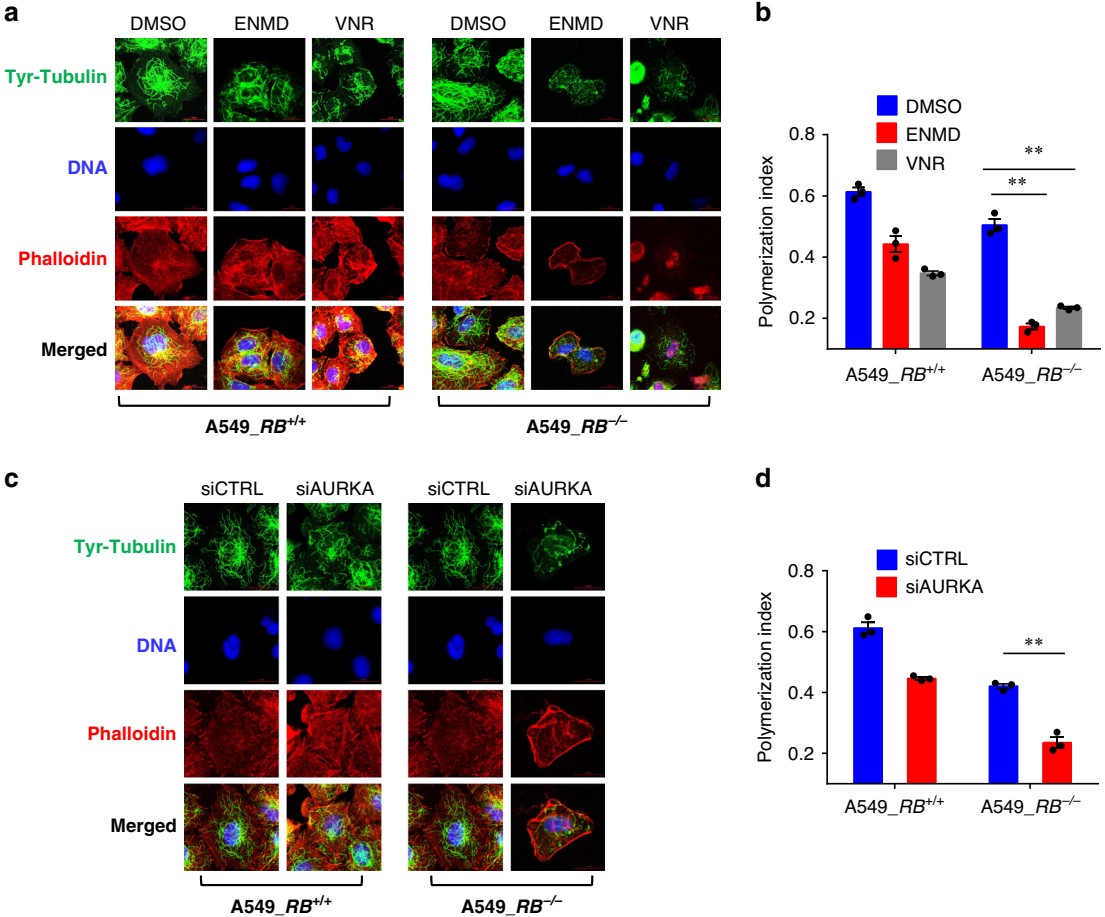

**Fig. 6 AURKA inhibitors destabilize microtubules in $RB1^{-/-}$ lung cancer cells. a, b** RB1-isogenic cells were treated with 2.5 μM ENMD-2076 (ENMD) or 25 nM vinorelbine (VNR) for 24 h, and the cells were processed for immunostaining of tyrosinated tubulin (Tyr-tubulin), DNA, and actin (phalloidin) (**a**). Scale bars, 20 μm. The total area of the cell defined by actin (phalloidin) staining was calculated and the ratio of microtubule polymers (defined by Tyr-tubulin) to cell area was determined as the polymerization index (**b**) as described in Methods. Data are presented as mean ± SEM ($n = 3$ independent experiments). **$P < 0.01$ between two indicated groups, determined using two-sided Student's $t$ test. **c, d** RB1-isogenic cells were treated with 25 nM AURKA siRNA (siAURKA) or control siRNA (siCTRL) for 48 h, and the cells were processed for immunostaining to determine microtubule polymerization index as described above. Scale bars, 20 μm. Data are presented as mean ± SEM ($n = 3$ independent experiments). **$P < 0.01$ between two indicated groups, determined using two-sided Student's $t$ test.

## Methods

**Cell culture, reagents, and antibodies**. The human lung cancer cell lines, A549, HCC827, NCI-H1650, NCI-H1975, NCI-H446, and NCI-H82 and the human breast cancer cell line, MDA-MB-468 were obtained from American Type Culture Collection (ATCC, Manassas, VA, USA). The human lung cancer cell lines were cultured in Roswell Park Memorial Institute (RPMI)-1640 medium supplemented with 10% fetal bovine serum (FBS) (Thermo Fisher Scientific, Waltham, MA) at 37 ℃ with 5% $CO_2$. MDA-MB-468 cells were maintained in Dulbecco's Modified Eagle's medium (DMEM) (Thermo Fisher Scientific) supplemented with 10% FBS at 37 ℃ with 5% $CO_2$. All the cell lines were regularly confirmed for free of mycoplasma by the iPSC Core facility in the Faculty of Health Sciences, University of Macau (https://fhs.umac.mo/research/ipsc-core/). ENMD-2076 (DC7118),

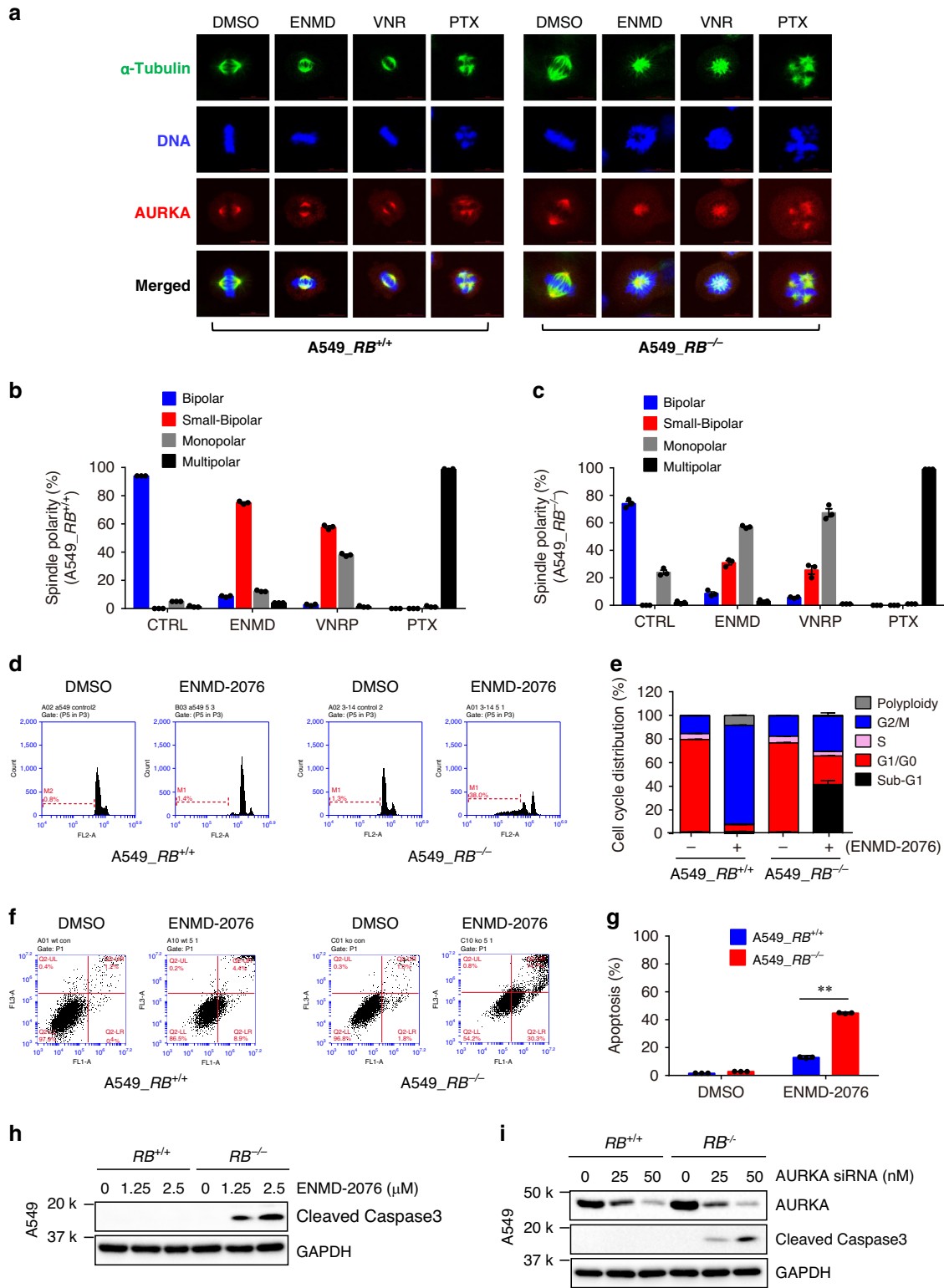

vinorelbine ditartrate (DC4184), and paclitaxel (DC8638) were purchased from DC Chemicals (Shanghai, China). siRNAs for AURKA (hs.Ri.AURKA.13.1), α-tubulin (hs.Ri.TUBA1A.13.1) and stathmin (hs.Ri.STMN1.13.1) were purchased from Integrated DNA Technologies (Singapore). Plasmid constructs, including Stathmin-GFP (Addgene #86782) and pEGFP-C1-FLAG (Addgene #46956) were gifts from Lynne Cassimeris[53] and Steve Jackson[54], respectively. Hoechst 33342 (H3570) and Rhodamine Phalloidin (R415) were purchased from Thermo Fisher Scientific. Imprint® Chromatin Immunoprecipitation Kit (CHP1) was purchased from Sigma-Aldrich (St. Louis, MO, USA). All the primary and secondary antibodies used in this study are listed in Supplementary Table 1.

**Generation of *RB1*$^{-/-}$ cell lines**. *RB1* CRISPR/Cas9 knockout (KO) plasmid (h, sc-400116) comprising a pool of three plasmids encoding green fluorescent protein (GFP) and three gRNAs targeting exon 7 and exon 8 of *RB1* gene (Supplementary Table 2) and *RB1* HDR plasmid (h, sc-400116-HDR) comprising a pool of two plasmids containing a homology-directed DNA repair (HDR) template with a dual selection marker (red fluorescent protein (RFP) and puromycin resistance gene) were purchased from Santa Cruz Biotechnology (Dallas, TX). *RB1* CRISPR/Cas9 KO plasmid and *RB1* HDR plasmid were cotransfected into A549 and HCC827 cells using Lipofectamine 3000 (Thermo Fisher Scientific). The *RB1*$^{-/-}$ clones were selected with RFP fluorescence and puromycin antibiotics (1 μg/mL).

**Fig. 7 AURKA inhibition induces abnormal spindle formation and apoptosis in $RB1^{-/-}$ lung cancer cells. a–c** Spindle morphology analysis of mitotic cells treated with AURKA inhibitor and microtubule polymerization inhibitors. A549 $RB1^{+/+}$ and $RB1^{-/-}$ cells were treated with 2.5 μM ENMD-2076 (ENMD), 25 nM vinorelbine (VNR) and 25 nM paclitaxel (PTX) for 24 h. Bortezomib (100 nM) was added at the last 2 h. **a** The spindle and mitotic DNA were analyzed with the immunofluorescence staining of α-tubulin (green), AURKA (red), and DNA (blue). Scale bars, 10 μm. **b, c** All the mitotic cells from the confocal images were analyzed for spindle polarity and quantitated based on four criteria: normal bipolar, small (short) bipolar, monopolar, and multipolar. Data are presented as mean ± SEM ($n = 3$ independent experiments. For each experiment, total 93–558 mitotic A549 $RB1^{+/+}$ cells (**b**) and total 88-580 mitotic A549 $RB1^{-/-}$ cells from each treatment condition were analyzed). **d, e** Effect of AURKA inhibitor on the cell cycle progression of A549 $RB1^{+/+}$ and $RB1^{-/-}$ cells. Cells were treated with 2.5 μM ENMD-2076 for 48 h and the cell cycle was analyzed by flow cytometry. $X$-axis represents propidium iodide and $Y$-axis represents cell count. Data are presented as mean ± SEM ($n = 3$ independent experiments). **f, g** Effect of AURKA inhibitor on apoptosis of A549 $RB1^{+/+}$ and $RB1^{-/-}$ cells. Cells were treated with 2.5 μM ENMD-2076 for 72 h and cell apoptosis was measured with Annexin V-FITC/propidium iodide staining. $X$-axis represents Annexin V-FITC and $Y$-axis represents propidium iodide. Data are presented as mean ± SEM ($n = 3$ independent experiments). \*\*$P < 0.01$ between two indicated groups, determined using two-sided Student's $t$ test. **h, i** Effect of AURKA inhibitor on apoptosis of A549 $RB1^{+/+}$ and $RB1^{-/-}$ cells. Cells were treated with the indicated concentrations of ENMD-2076 (**h**) or AURKA siRNA (**i**) for 72 h and cell apoptosis was measured with the Western blots of cleaved caspase-3.

---

$RB1^{-/-}$ clones were verified by Western blot, immunofluorescence, and Sanger sequencing with the primers designed to amplify the exon 7 and exon 8 of the $RB1$ gene containing the gRNA target sites (Supplementary Table 2). A primer targeting loxP site in HDR plasmid was used to confirm the HDR insertion in the $RB1$ KO clones (Supplementary Fig. 1f).

**Western blot analysis.** Cells were lysed with RIPA buffer (Thermo Fisher Scientific), containing phosphatase inhibitor and protease inhibitor cocktails (Roche Life Sciences, Mannheim, Germany). Protein concentration was measured using Pierce BCA Protein Assay Kit (Thermo Fisher Scientific). The protein lysates were boiled with SDS sample buffer at 95 °C for 5 min and then separated on SDS-PAGE. The separated proteins were transferred onto the nitrocellulose membrane and the membrane was blocked with 5% nonfat dried milk for 1 h at room temperature. The membrane was incubated with the indicated primary antibody overnight at 4 °C, followed by horseradish peroxidase-conjugated secondary antibodies for 1 h at room temperature. The protein bands were detected with the enhanced chemiluminescence solution (Thermo Fisher Scientific) under a ChemiDoc MP imaging system (Bio-Rad, Hercules, CA, USA). Image Lab (5.1 and 5.2.1) was used to acquire and analyze Western blot images.

**Epigenetics siRNA screening.** The epigenetics siRNA library (G-006105, 463 siRNAs containing a pool of 4 siRNAs targeting each of 463 epigenetics genes) used for the screening was purchased from GE Healthcare Dharmacon (Lafayette, CO). A549 $RB1^{+/+}$ and A549 $RB1^{-/-}$ cells were screened with the siRNA library in parallel with a reverse-transfection format in 384-well plates using Lipofectamine 3000 (Thermo Fisher Scientific) according to the manufacturer's instructions. The final concentration of the siRNAs was 50 nM. The negative control siRNA and the positive control siRNA (GAPDH siRNA) were included in each plate for the quality control of the siRNA screening. After 72 h of the transfection, the cell viability was measured using AlamarBlue® reagent. SoftMax Pro 6.3 was used to acquire the fluorescence of AlamarBlue® reagent. The gene silencing efficiency of the GAPDH siRNA in each 384-well plate was determined by the Western blot of the GAPDH protein from the whole cell lysates from the well. The screening was done in duplicate. The average cell viability ratios of $RB1^{-/-}/RB1^{+/+}$ for each siRNA were transformed into Z scores and plotted with GraphPad Prism 6.0 software (GraphPad Software, La Jolla, CA).

**Epigenetics drug screening.** The epigenetics compound library (L1900, 128 compounds) used for the screening was purchased from Selleck Chemicals (Houston, TX). A549 $RB1^{+/+}$ and A549 $RB1^{-/-}$ cells were screened in parallel with the library with the eight-dose, interplate-titration format in 384-well plates (Corning, #3656) using Liquidator-96 multi-well pipettor (Mettler Toledo, Columbus, OH). The final concentrations of the compound library was from 14 nM to 30 μM. After 72 h, the cell viability was measured using AlamarBlue® reagent (Thermo Fisher Scientific). The fluorescence of AlamarBlue® reagent (Ex560/Em590) was measured by a SpectraMax-M5 multi-well plate fluorescence reader (Molecular Devices, Sunnyvale, CA). The half-maximal inhibitory concentration ($IC_{50}$) was calculated with GraphPad Prism 6.0. The screening was done in duplicate and the average $IC_{50}$ values for each compound against the A549-RB1 isogenic cell pair were determined. The selectivity indices (SI) toward $RB1^{-/-}$ cells were calculated to identify synthetic lethality hits: $SI = IC_{50}^{RB1+/+}/IC_{50}^{RB1-/-}$.

**Cell cycle and apoptosis assays.** Cells treated with ENMD-2076 or transfected with AURKA siRNA were harvested, washed with PBS and then fixed with cold 70% ethanol overnight at −20 °C. The fixed cells were washed with PBS, suspended in 0.5 mL PBS, containing 50 μg mL$^{-1}$ propidium iodide (PI), 0.1 mg mL$^{-1}$ RNase A and 0.1% Triton X-100 for 30 min at 37 °C. Then, the cells were analyzed for cell cycle under a BD Accuri C6 flow cytometer (BD Biosciences, San Jose, CA). Cell

apoptosis was detected with FITC-Annexin V Apoptosis Detection Kit (BioLegend, San Diego, CA) using the BD Accuri C6 flow cytometer. BD Accuri C6 Software (1.0.264.21) was used to acquire and analyze cell cycle and apoptosis data.

**Reverse transcription-quantitative polymerase chain reaction (RT-qPCR).** Total RNA was extracted with RNeasy Mini Kit (Qiagen, Hilden, Germany). cDNA was synthesized by RT using High-Capacity cDNA Reverse Transcription Kit (Thermo Fisher Scientific) according to the manufacturer's instructions. The mRNA expression of TUBA1A, STMN1, STMN3, EML2, TPPP, TPPP3, NCALD, CLIP1, IFT81, PARK2, and LRRK2 was detected with qPCR using iTaq Universal SYBR Green Supermix (Bio-Rad, Hercules, CA) using an ABI-7500 Real-Time PCR System (Thermo Fisher Scientific) with the primer pairs (Supplementary Table 2). 7500 software (v2.3) was used to acquire and analyze qPCR data. Glyceraldehyde-3-phosphate dehydrogenase (GAPDH) was used as the internal control. Relative mRNA level was quantitated using the comparative CT ($2^{-\Delta\Delta Ct}$) method.

**Tumor xenograft mouse models.** All animal procedures were approved by the Animal Research Ethics Committee of the University of Macau. All animals were maintained in the University of Macau, specific pathogen free (SPF) Animal Facility. Six- to eight-week female athymic nude mice (The Jackson Laboratory, Bar Harbor, ME) were used for tumor xenografts. Mice were housed in a fully climate-controlled room at constant temperature and humidity on a 12:12 h light/dark cycle with free access to food and water. A549 $RB1^{+/+}$ cells ($5 \times 10^6$ cells/mouse), A549 $RB1^{-/-}$ cells ($1 \times 10^7$ cells/mouse), HCC827 $RB1^{+/+}$ cells ($5 \times 10^6$ cells/mouse) and HCC827 $RB1^{-/-}$ cells ($1 \times 10^7$ cells/mouse) suspended in Matrigel were implanted subcutaneously into nude mice. In each mouse, isogenic $RB1^{+/+}$, and $RB1^{-/-}$ cells were implanted bilaterally. After tumors were palpable, the mice were treated with vehicle (sterile saline, containing 5% DMSO, 5% tween-80 and 5% polyethylene glycol-400, daily), ENMD-2076 (25 and 50 mg kg$^{-1}$, daily), alisertib (25 and 50 mg kg$^{-1}$, daily) or Aurora A Inhibitor I (25 and 50 mg kg$^{-1}$, daily) for 33 days (A549 tumor mice) and 36 days (HCC827 tumor mice) via i.p. injection. Tumor size was measured periodically by calipers and the tumor volume (mm³) was determined by the ellipsoid formula (volume = L × W² × π/6). Mouse body weight was measured regularly to assess drug toxicity. For Western blots for tumor samples, tumor tissues were cut into pieces and placed in a pre-cooled 5 mL tube filled with ice-cold RIPA buffer with protease inhibitor and phosphatase inhibitor cocktails. The tumors were then homogenized with Polytron® PT 1200 E Manual Disperser (Kinematica, Bohemia, NY) on ice and the lysate was transferred to precooled 1.5 mL tube for centrifugation at $13,523 \times g$ for 15 min at 4 °C. The supernatants containing tissue proteins were collected and measured for protein concentration. Approximately each 40 μg of protein sample was run on a SDS-PAGE and analyzed for Western blots. For immunofluorescence analysis of tumor tissues, tumor samples were embedded in OCT (optimum cutting temperature) compound (Sakura Finetek, Alphen aan den Rijn, Netherlands) and sectioned into 10 μm with Leica CM3050 S Cryostat (Leica Biosystems, Wetzlar, Germany). The tumor sections were fixed in cold acetone at −20 °C for 20 min, permeabilized with 0.5% Triton X-100 for 20 min and washed with PBS prior to blocking in 3% BSA in PBS containing 0.1% Tween 20 for 1 h. The tumor sections were then incubated with anti-Ki67 at 4 °C overnight followed by incubation with Hoechst33342 (Thermo Fisher Scientific) and secondary antibodies conjugated with Alexa Fluor-488 for 1 h at room temperature. The samples were washed with PBS, mounted with Immu-mount (Thermo Fisher Scientific), and observed under a Carl Zeiss LSM 710 confocal microscope (Carl Zeiss, Thornwood, NY).

**RNA-sequencing analysis.** Total RNA from A549 $RB1^{+/+}$ and $RB1^{-/-}$ cells was extracted with RNeasy Kit (74136, Qiagen, Germany) and cDNA library was constructed with NEBNext® Ultra™ Directional RNA Library Prep Kit for Illumina® (NEB #E7760, New England Biolabs, Ipswich, MA, USA). cDNA library was sequenced on an Illumina Hi-Seq (Illumina, San Diego, CA) in the Genomics and

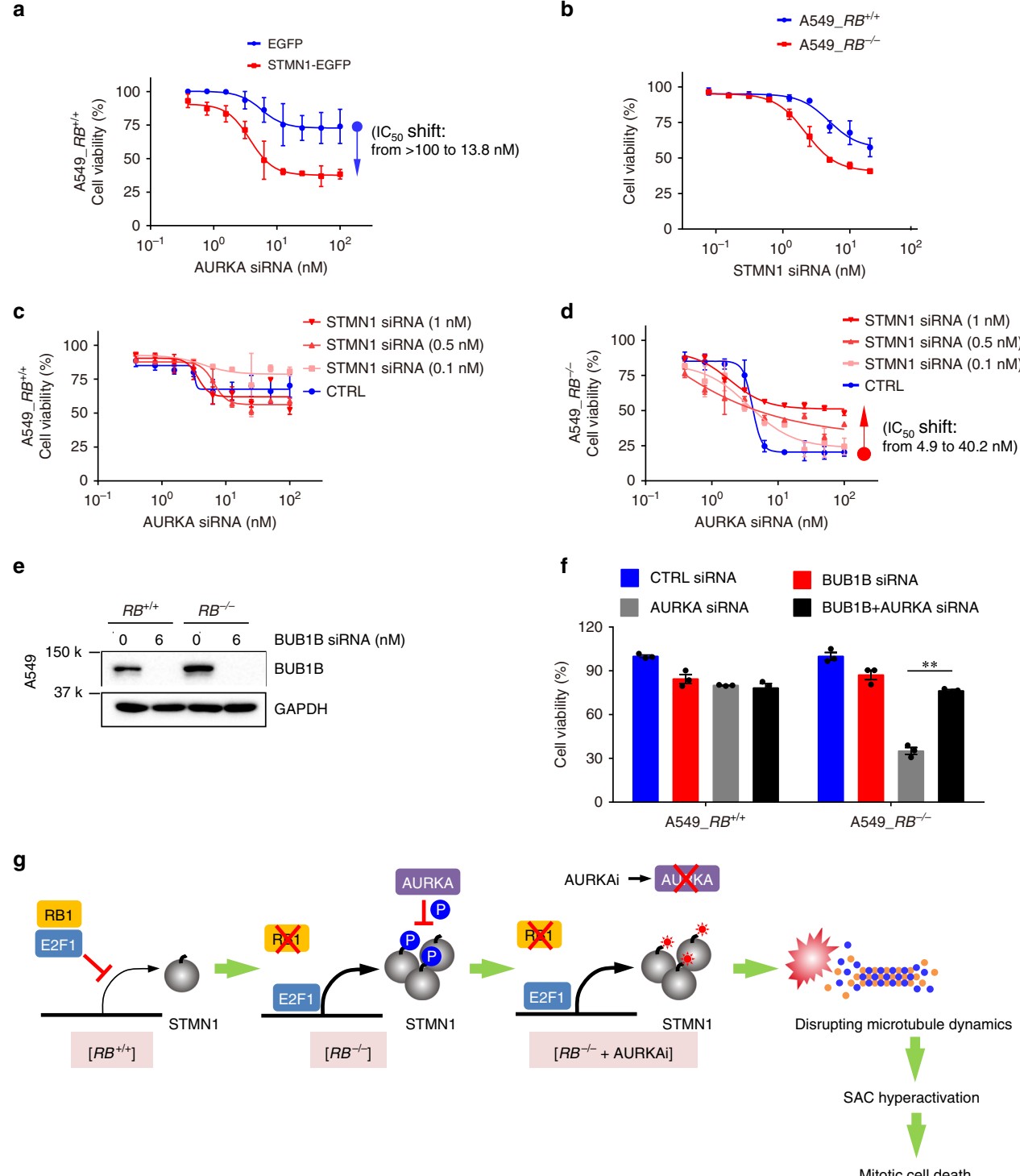

**Fig. 8 Stathmin mediates the synthetic lethal effect of AURKA inhibition. a** Effect of stathmin overexpression on the sensitivity of $RB1^{+/+}$ cells to AURKA inhibition. A549 $RB1^{+/+}$ cells were transfected with 1 μg EGFP control vector or EGFP-STMN1 overexpression vector and the cell $IC_{50}$ values for the AURKA siRNA were determined with AlamarBlue assay. The changes in $IC_{50}$ values are indicated. **b** Effect of stathmin siRNA on the cell viability of A549 $RB1^{+/+}$ and $RB1^{-/-}$ cells. Effect of stathmin siRNA on the sensitivity of $RB1^{+/+}$ (**c**) and $RB1^{-/-}$ (**d**) cells to AURKA inhibition. Cells were transfected with control siRNA (CTRL) or three nonlethal concentrations of stathmin siRNA (0.1, 0.5, and 1 nM) and the cell $IC_{50}$ values for the AURKA siRNA were determined with AlamarBlue assay. The changes in $IC_{50}$ values are indicated. Data are presented as mean ± SD ($n = 3$ independent experiments). **e, f** Effect of BUB1B siRNA on the sensitivity of $RB1^{+/+}$ and $RB1^{-/-}$ A549 cells to AURKA silencing. Cells were transfected with control siRNA (CTRL), 6 nM BUB1B siRNA, 25 nM AURKA siRNA or siRNA combination for 72 h, and the BUB1B silencing efficiency (**e**) and cell viability (**f**) were determined. Data are presented as mean ± SEM ($n = 3$ independent experiments). **P < 0.01 between two indicated groups, determined using two-sided Student's $t$ test. **g** Proposed working model of the synthetic lethality between RB1 and AURKA.

Bioinformatics Core Facility of Faculty of Health Sciences, University of Macau. Raw RNA-Seq data was subjected to FastQC quality control. Quality confirmed RNA-seq reads from each library were aligned and mapped with TopHat 2.1.1 (Center for Computational Biology at Johns Hopkins University)[55]. Expression quantification was defined with FPKM (Fragments Per Kilobase of transcript per Million mapped reads) by Cufflinks 2.2.1 (Trapnell-Lab, Github) after mapping. Filtered gene sets were identified to be upregulated or down-regulated 1.5-fold in paired samples. For the functional annotation of the transcriptome profiles, genes showing differential expression was subjected to the GO analysis with the web-based tool, DAVID v6.8 (The Database for Annotation, Visualization, and Integrated Discovery, National Institute of Allergy and Infectious Diseases, NIH).

**Microtubule stability assay.** Microtubule stability assay was performed based on the previous report[56]. Cells were treated with or without drugs for 24 h, or transfected with or without siRNA for 48 h. Cells were then suspended in the extraction buffer, containing 0.1 M PIPES, pH 7.1, 1 mM $MgSO_4$, 1 mM EGTA, 2 M glycerol, 0.1% Triton X-100, and protease inhibitor cocktail. After incubation on ice for 15 min, the cell lysates were centrifuged at $21,130 \times g$ for 15 min at 4 °C. The supernatant, containing 0.1% Triton-soluble tubulins, was collected. The remaining pellet was resuspended in the lysis buffer, containing 25 mM Tris-HCl, pH 7.4, 0.4 M NaCl, and 0.5% SDS, and boiled for 10 min. The sample was centrifuged at $21,130 \times g$ for 5 min, and the polymeric tubulin-containing supernatant was collected. The soluble (S) and polymerized (P) tubulin fractions were subjected to SDS-PAGE and detected by Western blot with anti-α-tubulin antibody. An anti-GAPDH antibody was used as a loading control for each experimental condition. The microtubule polymerization status in each experimental condition was quantitated based on the relative amount of P out of total (P + S).

**Immunofluorescence analyses for microtubule polymerization and spindle polarity.** For the measurement of microtubule polymerization status, A549 $RB1^{+/+}$ and $RB1^{-/-}$ cells seeded in Nunc Lab-Tek II 8-Chamber Slide (Thermo Fisher Scientific) were treated with ENMD-2076 or microtubule antagonists for 24 h. For the spindle polarity measurement, A549 $RB1^{+/+}$ and $RB1^{-/-}$ cells treated with ENMD-2076 or microtubule antagonists for 24 h were further treated with bortezomib (100 nM) at 2 h prior to the cell fixation to arrest mitotic cells at the metaphase-to-anaphase transition[57]. The cells were fixed with 4% paraformaldehyde and permeabilized with 0.5% Triton X-100 in PBS. After being blocked in 3% BSA in PBS, the cells were incubated with primary antibodies against α-tubulin, Tyr-tubulin, and/or AURKA overnight at 4 °C. Cells were incubated with secondary antibodies conjugated with Alexa Fluor-488 for 1 h at room temperature. The cellular actin was stained with rhodamine-phalloidin (R415,Thermo Fisher Scientific) for 1 h at room temperature and the cellular nuclei were stained with Hoechst 33342 for 5 min. Then the cells were mounted with Immu-mount and observed under a Carl Zeiss LSM 710 confocal microscope. ZEN2012 (black version) was used to acquire and analyze confocal images. The microtubule polymerization index was calculated with Tubeness plugin for ImageJ, which has been developed by Harkcom et al.[42]. The total area of the cell defined by actin (rhodamine-phalloidin) staining was calculated and the ratio of microtubule polymers to cell area was determined as the polymerization index. For the quantitation of spindle polarity, abnormal spindles were defined as those that do not display normal bipolar spindle formation, as defined by the existence of a clearly visible metaphase plate straddled by undisrupted radial arrays of microtubules emanating from opposite poles[57].

**Chromatin immunoprecipitation (ChIP)-quantitative polymerase chain reaction (qPCR) assay.** Chromatin immunoprecipitation (ChIP) was performed using an Imprint Chromatin Immunoprecipitation Kit (CHP1, Sigma-Aldrich) following the manufacturer's instructions. Cells were cross-linked in 1% formaldehyde and neutralized using 0.125 M glycine. The chromatin complex was sonicated using Bioruptor Sonication System (Diagenode, Denville, NJ). The sheared chromatin was immunoprecipitated with anti-E2F1 (10 μL per sample for ChIP), anti-E2F2 (4 μg per sample for ChIP), and anti-E2F3 (4 μg per sample for ChIP) ChIP-grade antibodies using the assay wells pre-coated with protein A. The normal mouse or rabbit IgG was used as a non-specific antibody control for immunoprecipitation. Protein–DNA cross-linking was reversed at 65 °C. DNA was purified and analyzed by real-time qPCR. Enrichment of DNA was shown as the percentage (%) input according to the manufacturer's instruction. The calculation of the % Input for each ChIP fraction is $2^{(-\Delta Ct \ [\text{normalized ChIP}])}$, where ΔCt [normalized ChIP] = (Ct [ChIP] - (Ct [Input] - Log2(Input Dilution Factor))). Primers used in ChIP-qPCR analysis are listed in Supplementary Table 2.

**Analysis of ChIP-Atlas database.** ChIP-Atlas database (http://chip-atlas.org/) was used to predict the E2F1 target genes among the tubulin binding proteins. Target genes were accepted if the peak-call intervals of E2F1 overlapped with a transcription start site ±10 kb. The E2F1 binding score was calculated based on MACS2 scores of peak-call data and STRING, a comprehensive database recording protein-gene interactions based on experimental evidence.

**Statistics and reproducibility.** Statistical differences of the data between control and test groups were determined by two-sided Student's $t$ test or one sample $t$ test using Graphpad Prism 6.0. $P$ values < 0.05 was considered significant. All the Western blot data shown are representative data from at least three independent experiments.

**Reporting summary.** Further information on research design is available in the Nature Research Reporting Summary linked to this article.

## Data availability

The RNA-Seq data have been deposited in the NCBI's Gene Expression Omnibus (GEO) database under the accession code GSE132722. The E2F1 ChIP-Seq data referenced during the study are available in a public repository from the ChIP-Atlas database (http://chip-atlas.org/). The proteomics data from 365 lung cancer patients (TCGA-LUAD cohort) are available in The Cancer Genome Atlas (TCGA) project (https://www.cancer.gov/about-nci/organization/ccg/research/structural-genomics/tcga) and are archived in LinkedOmics (http://www.linkedomics.org/) for expression analysis. All the other data supporting the findings of this study are available within the article and its Supplementary Information files and from the corresponding author upon reasonable request.

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

## Acknowledgements

We thank to the members of the Genomics and Bioinformatics Core of FHS and FHS Animal Facility at the University of Macau for experimental and technical supports. Transcriptome profiling work was performed in part at the High Performance Computing Cluster (HPCC) which is supported by Information and Communication Technology Office (ICTO) of the University of Macau. This study was supported by the Multi-Year Research Grants (MYRG2017-00176-FHS and MYRG2019-00116-FHS) to J.S.S. from the University of Macau.

## Author contributions

Conceptualization and study design: J.L. and J.S.S.; development of methodology: J.L. and E.J.Y.; acquisition of data: J.L., E.J.Y., B.Z., C.W., L.P., C.S., P.K.M., Y.L., and K.T.; analysis and interpretation of data: J.L., B.Z., C.W., L.P., C.S., P.K.M., Y.L., K.T., and J.S.S.; writing, reviewing, and revision of the manuscript: J.L. and J.S.S.; study supervision and funding acquisition: J.S.S.

## Competing interests

The authors declare no competing interests.
