## [Peer Review File · Nature Communications]

Reviewers' comments:

Reviewer #1 (Remarks to the Author):

Review of Nature Communications 19-39594

In this manuscript the authors undertake siRNA and small molecule screens to detect selective sensitivities of RB1 deficient cells. They generate isogenic cell lines using CRISPR deletions of RB1 and carry out screens in both to determine RB1 dependencies. This identified Aurora Kinase A inhibitors and Aurora Kinase A depletion as strong synthetic interactions. The authors then proceed to determine that Tubulin expression is diminished in RB1 knock out cells and that Stathmin, a microtubule destabilizing protein is essential for this phenotype. Beyond the work that solidifies the importance of Aurora Kinase inhibitors this work also demonstrates important new insights into the mechanism(s) to exploit RB1 deficiency. Thus the value of this work to a broad readership of scientists is obvious. However, there are some shortcomings that should be addressed for publication.

- 1) The biggest shortcoming is the conceptualization of the role of Stathmin and microtubule dynamics as a separate consequence from SAC elevation in response to RB1 deletion. Indeed, the authors describe it as a second model of vulnerability caused by RB1 deficiency in the discussion. It isn't clear to me that they have to be separate, only that the authors describe it this way. A provocative publication not included by the authors is Kabeche et al. Current Biology 2012 that argues that SAC overactivation can stabilize kinetochore-microtubule attachments leading to defects in mitosis. Based on this I'm not convinced the observations here are independent of SAC overactivation in RB1 deficiency. The authors should investigate if SAC signaling levels are elevated in their RB1 knock out cells and if reducing them with partial siRNA depletion (that they use for many genes) can ameliorate the synthetic lethality that they observe. New data in this area would substantiate if microtubule regulation is indeed separate from the SAC or if these phenotypes are linked.
- 2) The authors also propose that Stathmin is upregulated as a consequence of E2F1 activity that is liberated in RB1 deletion scenarios. Given that E2F1 can bind almost anywhere in the genome, the ChIP experiments that are used to argue this point in figure 4 are inadequate. The data mining from other ChIP sequence or cancer proteomic databases is intriguing but does not replace demonstration of this by experimentation. The ChIP methods used are undescribed in the methods section and the 'fold-enrichment' calculation in Fig. 4i is unclear. Fold-enrichment is usually relative to something like an IgG control, so showing it in this graph doesn't make sense. To demonstrate specificity in E2F1 occupation of this promoter, akin to other cell cycle regulatory targets of E2Fs is to analyze chromatin occupancy at a neutral genomic location (eg. 10 kb 5' of Stathmin TSS). Then quantitate as percent chromatin precipitated by IgG and E2F1 antibodies at both locations to determine if there is enrichment at the proximal promoter relative to background. This type of analysis at a minimum is necessary to establish if this is a bona fide transcriptional target. The authors might also consider checking occupancy by other E2F activator family members to be sure if it is actually an E2F1 target gene instead of an more general target of E2Fs as a family.
- 3) The quantitation of microtubule polymerization by pelleting and western blotting is concerning as the differences reported in the graphs are small and the normalization of comparators is unclear as this technique is not described in the methods. I assume that the authors rely on GAPDH protein levels to normalize the amounts of Tubulin precipitated by centrifugation. To use a contaminating soluble protein as a loading control is risky. In addition, the authors show that Aurora Kinase A siRNA depletion ablated polymerization in RB1^{-/-} A549 cells, however, there is no evidence from intact cells that siRNA depletion prevents spindle formation and none of the conditions shown in Figure 6 block microtubule formation. These experiments need to be better explained and their conclusions need to be supported by separate experiments that use in vivo measurements of microtubule levels to validate their in vitro measurements.

Reviewer #2 (Remarks to the Author):

In this manuscript it was investigated, using chemical and genetic screens, dependencies of novel targets in RB1-deficient cells and Aurora A was identified. They present evidence that RB1^{-/-} cells have unbalanced microtubule dynamics in which stathmin, an E2F transcriptional target, is involved. Aurora A phosphorylation of stathmin facilitates the microtubule destabilization and rescue experiments showed that stathmin depletion reversed the synthetic lethality by Aurora A. All together this paper is interesting. However, authors do not present a complete package of data/evidence to strongly support the novelty of the Aurora A selective dependency of RB-deficient cancer cells.

1. It is important to provide evidence for the proliferation rate of the isogenic models. A small difference in this will affect the hit identification, especially if the library is including cell cycle interfering inhibitors.
2. It needs to be more convincing explanation of why do Aurora inhibitors are included in the epigenetic library. There are dozens of kinases that are regulating transcription factors.
3. Other mitotic inhibitors such as TTK, PLK1, Eg5 need to be used to exclude a general mitotic kinases effect.
4. The major issue for the project is the use of ENDM-2076 at all the experiments throughout the study and the manuscript. ENDM-2076 is not a selective Aurora A inhibitor. Is a multi-kinase inhibitor including Aurora A, Aurora B, FGFR, FLT3, c-kit, VEGFR and many more. How does only one compound, moreover, not the right one, proved the case?
5. The authors should use more selective Aurora A inhibitors such as alisertib throughout the experiments described in Figures: 1i, k, 2a-j, 3a, 5ac,d,e, 6a,d,f,g,h.

In conclusion, I do have significant concerns that need to be addressed before recommending it for publication in Nat Communications.

Author Responses to Referees

Reviewer #1 (Remarks to the Author):

In this manuscript the authors undertake siRNA and small molecule screens to detect selective sensitivities of RB1 deficient cells. They generate isogenic cell lines using CRISPR deletions of RB1 and carryout screens in both to determine RB1 dependencies. This identified Aurora Kinase A inhibitors and Aurora Kinase A depletion as strong synthetic interactions. The authors then proceed to determine that Tubulin expression is diminished in RB1 knock out cells and that Stathmin, a microtubule destabilizing protein is essential for this phenotype. Beyond the work that solidifies the importance of Aurora Kinase inhibitors this work also demonstrates important new insights into the mechanism(s) to exploit RB1 deficiency. Thus the value of this work to a broad readership of scientists is obvious. However, there are some shortcomings that should be addressed for publication.

Response: We are grateful to the reviewer for his/her time to read our manuscript and providing all the helpful comments and suggestions. We tried to fully address the reviewer's concerns by including several additional experimental data and modifying texts based on the reviewer's comments. Described below are point-to-point responses to the reviewer's specific comments. All the text revisions are marked in red in the revised manuscript.

1) The biggest shortcoming is the conceptualization of the role of Stathmin and microtubule dynamics as a separate consequence from SAC elevation in response to RB1 deletion. Indeed, the authors describe it as a second model of vulnerability caused by RB1 deficiency in the discussion. It isn't clear to me that they have to be separate, only that the authors describe it this way. A provocative publication not included by the authors is Kabeche et al. Current Biology 2012 that argues that SAC overactivation can stabilize kinetochore-microtubule attachments leading to defects in mitosis. Based on this I'm not convinced the observations here are independent of SAC overactivation in RB1 deficiency. The authors should investigate if SAC signaling levels are elevated in their RB1 knock out cells and if reducing them with partial siRNA depletion (that they use for many genes) can ameliorate the synthetic lethality that they observe. New data in this area would substantiate if microtubule regulation is indeed separate from the SAC or if these phenotypes are linked.

Response: We totally agree to the reviewer that stathmin/microtubule dynamics and SAC are linked, and are not separate phenotypes. As we have shown in the model (Fig. 7e in the original submission), as well as described in the later part of discussion in

the original submission, stathmin-mediated disruption of microtubule dynamics will eventually lead to SAC elevation in RB1-deficient cells. However, as the reviewer pointed out, what we described in the original submission overall was not exactly the way we intended. We therefore modified discussion section to the way that the two phenotypes are closely linked in inducing synthetic lethality (marked in red in the discussion section of the revised manuscript). We also investigated SAC signaling levels and conducted rescue experiment with siRNA depletion of the key SAC component BUB1B in RB1 KO cells treated with AURKA siRNA. Our data support the involvement of SAC activation in RB1 KO cells following the disruption of the microtubule dynamics by AURKA inhibition (Figure 8e and f) in the revised manuscript).

Revised Figure 8 (e-g). **e**, The level of BUB1B is elevated in *RB*^{-/-} cells. **f**, BUB1B silencing significantly rescues the synthetic lethal effect of AURKA siRNA in *RB*^{-/-} cells. **g**, Working model: [1] In *RB*^{+/+} cells, stathmin expression is tightly regulated by the RB1/E2F complex. In *RB*^{-/-} cells, activated E2F upregulates stathmin expression. [2] The overexpression of stathmin in *RB*^{-/-} cells facilitates microtubule depolymerization in cells, such that the cells have unbalanced microtubule dynamics. This unbalance is not likely to be critical for the cell viability in *RB*^{-/-} cells because AURKA phosphorylates stathmin and partially suppresses its activity on microtubule stability. [3] When AURKA activity is inhibited, AURKA no longer phosphorylates and suppresses stathmin, thereby activating the stathmin proteins that have been overexpressed and heavily disrupting the microtubule dynamics in *RB*^{-/-} cells. [4] The disruption of microtubule dynamics triggers SAC hyperactivation in *RB*^{-/-} cells, leading to mitotic cell death.

New Figures were added and relevant texts were revised in the revised manuscript.

2) The authors also propose that Stathmin is upregulated as a consequence of E2F1 activity that is liberated in RB1 deletion scenarios. Given that E2F1 can bind almost anywhere in the genome, the ChIP experiments that are used to argue this point in figure 4 are inadequate. The data mining from other ChIP sequence or cancer proteomic databases is intriguing but does not replace demonstration of this by experimentation. The ChIP methods used are undescribed in the methods section and the ‘fold-enrichment’ calculation in Fig. 4i is unclear. Fold-enrichment is usually relative to something like an IgG control, so showing it in this graph doesn’t make sense. To demonstrate specificity in E2F1 occupation of this promoter, akin to other cell cycle regulatory targets of E2Fs is to analyze chromatin occupancy at a neutral genomic location (eg. 10 kb 5’ of Stathmin TSS). Then quantitate as percent chromatin precipitated by IgG and E2F1 antibodies at both locations to determine if there is enrichment at the proximal promoter relative to background. This type of analysis at a minimum is necessary to establish if this is a bona fide transcriptional target. The authors might also consider checking occupancy by other E2F activator family members to be sure if it is actually an E2F1 target gene instead of an more general target of E2Fs as a family.

Response: We are grateful to the reviewer to point out this important issue. As per the reviewer’s suggestion, we used four different sets of primers, three targeting stathmin proximal promoter where predicted E2F binding sites are located and one targeting a neutral region (10 kb 5’ upstream of the TSS), and conducted ChIP experiments with E2F1, 2 and 3 antibodies. The resulting data were quantitated as percent chromatin precipitated by IgG and E2Fs antibodies. Our results show that E2F1, 2 and 3 all significantly bind onto the stathmin promoter, but not onto the neutral region (Fig. 4h-j in the revised manuscript). These data suggested that stathmin is a transcription target of E2F family transcription factors.

Revised Fig. 4h-j, ChIP-qPCR assay for stathmin promoter.

New Figures were added and relevant texts were revised in the revised manuscript.

3) The quantitation of microtubule polymerization by pelleting and western blotting is concerning as the differences reported in the graphs are small and the normalization of comparators is unclear as this technique is not described in the methods. I assume that the authors rely on GAPDH protein levels to normalize the amounts of Tubulin precipitated by centrifugation. To use a contaminating soluble protein as a loading control is risky. In addition, the authors show that Aurora Kinase A siRNA depletion ablated polymerization in RB1^{-/-} A549 cells, however, there is no evidence from intact cells that siRNA depletion prevents spindle formation and none of the conditions shown in Figure 6 block microtubule formation. These experiments need to be better explained and their conclusions need to be supported by separate experiments that use in vivo measurements of microtubule levels to validate their in vitro measurements.

Response: We are grateful to the reviewer to point out the quantitation issue of microtubule polymerization. First, we actually did not normalize the tubulin level in each lane. We used GAPDH protein levels as a loading control to visualize that we used an equal amount of cell lysate for microtubule polymerization assay in each experimental condition (DMSO, ENMD, Vinorelbine, etc). The cell lysate in each experimental condition was divided into two fractions, P (polymerized) and S (soluble). The quantitation for microtubule polymerization status was done only based on the relative amount of P out of total (P+S) in each experimental condition. We added the experimental details including the quantitation in the Methods section in the revised manuscript.

It is indeed a very good suggestion to provide in cell measurement of microtubule polymerization status. In Figure 6 in the original submission (Fig. 7 in the revised manuscript), we observed that spindle was much shorter in length and its structure was disorganized in AURKA inhibitor- or vinorelbine (a positive control compound as microtubule destabilizing agent)-treated cells. We also tested AURKA siRNA on spindle morphology and observed very similar phenotype changes, including small-bipolar in RB1 wildtype, and monopolar in RB1 KO cells (Supplementary Fig. 10a-d in the revised manuscript, also shown below).

Revised Supplementary Fig. 10a-d, Effect of AURKA siRNA on spindle morphology and polarity.

To further visualize the effect of AURKA inhibition on microtubule polymerization *in vivo*, we adopted a method developed by Harkcom et al (PNAS, 2014, 111(24):E2443-52) that used anti-Tyr-tubulin immunostaining to measure and quantitate the microtubule polymerization index in cells. Our result show that AURKA inhibitor, as well as siRNA silencing, significantly reduced microtubule polymerization index, a phenotype similar to that seen in vinorelbine treatment (Figure 6a-d in the revised manuscript, also shown below) or vinblastine (another microtubule depolymerizing agent; Harkcom et al. PNAS, 2014, 111(24):E2443-52) treatment.

Revised Fig. 6 (a-d). Anti-tyr-tubulin immunostaining to detect microtubule polymerization status in cells.

All new Figures were added and relevant texts were revised in the revised manuscript.

We greatly appreciate the reviewer again for thoughtful and constructive comments to improve the quality of our study. We hope that the revised version of the manuscript has been significantly improved to meet the standard of the reviewer and the journal.

Reviewer #2 (Remarks to the Author):

In this manuscript it was investigated, using chemical and genetic screens, dependencies of novel targets in RB1-deficient cells and Aurora A was identified. They present evidence that RB1^{-/-} cells have unbalanced microtubule dynamics in which stathmin, an E2F transcriptional target, is involved. Aurora A phosphorylation of stathmin facilitates the microtubule destabilization and rescue experiments showed that stathmin depletion reversed the synthetic lethality by Aurora A. All together this paper is interesting. However, authors do not present a complete package of data/evidence to strongly support the novelty of the Aurora A selective dependency of RB-deficient cancer cells.

Response: We appreciate the reviewer for all the helpful comments and suggestions. We conducted a number of experiments and revised the texts to fully address the reviewer's concerns. Described below are point-to-point responses to the reviewer's specific comments. All the text revisions are marked in red in the revised manuscript.

1. It is important to provide evidence for the proliferation rate of the isogenic models. A small difference in this will affect the hit identification, especially if the library is including cell cycle interfering inhibitors.

Response: As per the reviewer's suggestion, we analyzed the proliferation rate of A549 and HCC827 RB1-isogenic cell pairs by the real-time monitoring of cell proliferation using an IncuCyte. As shown in Supplementary Figure 2a and b, there is no significant difference in the proliferation rate between RB1 wildtype and KO cells. This information was described in the revised text (in the first section of Results), and the data and relevant experimental details are added to Supplementary Fig. 2 in the revised manuscript.

Revised Supplementary Fig. 2a and b. Real-time measurement of cell proliferation rates of RB1-isogenic lung cancer cell pairs used in this study.

2. It needs to be more convincing explanation of why do Aurora inhibitors are included in the epigenetic library. There are dozens of kinases that are regulating transcription factors.

Response: We are grateful to the reviewer to point out that this basic information was missed out from the manuscript. As the reviewer mentioned, several kinase inhibitors are included in the epigenetics compound library, including inhibitors of AURKA, JAK, CDK, and PIM. They are known to phosphorylate heterochromatin proteins, histones and/or EZH2 epigenetic writer, and regulate epigenetic dynamics. AURKA is known to phosphorylate a well known epigenetic regulator, heterochromatin protein 1 γ (HP1 γ) at Ser83, and regulate G2/M gene expression networks (Grzenda et al, 2013, Epigenetics & Chromatin, 6(1):21). It is also known to phosphorylate histone H3 at Thr 118 to regulate chromatin structure (Wike et al, 2016, Elife, 5:e11402) We included this information in the revised manuscript (in the first section of Results).

3. Other mitotic inhibitors such as TTK, PLK1, Eg5 need to be used to exclude a general mitotic kinases effect.

Response: As per the reviewer's suggestion, we tested BAY1217389, an inhibitor of TTK/Mps1, BI-2536, an inhibitor of PLK1, and Ispinesib, an inhibitor of Eg5 in the RB1 isogenic cell pair. We observed that, unlike AURKA inhibitors, these mitotic inhibitors did not exhibit significant synthetic lethal effect in RB1-deficient lung cancer cells. These data suggested that the synthetic lethality by AURKA inhibitors was not due to the general mitotic kinase inhibitory effect. The data were added to Supplementary Fig. 3a-c and the relevant description was added to the first section of Results.

Revised Supplementary Fig. 3a-c. Effect of mitotic inhibitors, including BAY1217389 (TTK/Mps1 inhibitor), BI-2536 (PLK1 inhibitor), and Ispinesib (Eg5 inhibitor) on the viability of RB1 isogenic cell pair.

4. The major issue for the project is the use of ENDM-2076 at all the experiments throughout the study and the manuscript. ENDM-2076 is not a selective Aurora A inhibitor. Is a multi-kinase inhibitor including Aurora A, Aurora B, FGFR, FLT3, c-kit, VEGFR and many more. How does only one compound, moreover, not the right one, proved the case?

Response: this comment is addressed together with the comment #5 below.

5. The authors should use more selective Aurora A inhibitors such as alisertib throughout the experiments described in Figures: 1i, k, 2a-j, 3a, 5ac,d,e, 6a,d,f,g,h.

Response: We are grateful to the reviewer to point out this important issue. We indeed agree to the reviewer that ENMD-2076 may not be the best choice as the AURKA inhibitor in this study. The very first reason we used ENMD-2076 throughout the study was that ENMD-2076 was the best synthetic lethal hit among the AURKA inhibitors identified from the screening (see Fig. 1g, selectivity index values. This data has been revised to clarify the drug names and targets in the revised Figure). In addition, we used AURKA siRNA in most of in vitro experiments in parallel with ENMD-2076 to cross validate that the synthetic lethal phenotype was due to the AURKA inhibition. However, as the reviewer's suggestion of using a more selective AURKA inhibitor makes much sense, we tested two additional AURKA inhibitors, including alisertib and Aurora A Inhibitor I (TC-S 7010), both of which are highly selective inhibitors of AURKA, for all in vitro and in vivo experiments. (Alisertib is 200-fold more selective (Manfredi et al, Clin Cancer Res. 2011;17(24):7614-24), and Aurora A Inhibitor I is 1000-fold more selective (Aliagas-Martin et al, J Med Chem. 2009; 52(10):3300–3307) for AURKA than AURKB). Shown below are all the new data that the reviewer suggested to repeat with alisertib and Aurora A Inhibitor I.

Revised Fig. 1j and m; Supplementary Fig. 2f and g, Effect of Alisertib and Aurora A Inhibitor I on the synthetic lethality *in vitro*.

Revised Fig. 2g-i; Revised supplementary Fig. 4a-i, Effect of Alisertib and Aurora A Inhibitor I on the synthetic lethality *in vivo*.

Revised supplementary Fig. 5a-h, Effect of Alisertib and Aurora A Inhibitor I on the synthetic lethality *in vivo*.

Revised supplementary Fig. 6c-d, Effect of Alisertib and Aurora A Inhibitor I on the mice body weight change.

Revised supplementary Fig. 7c-d, Effect of Alisertib and Aurora A Inhibitor I on α -tubulin level.

Revised supplementary Fig. 8a and b, Effect of Alisertib and Aurora A Inhibitor I on stathmin phosphorylation.

Revised supplementary Fig. 9c-f, Effect of Alisertib and Aurora A Inhibitor I on microtubule polymerization.

Revised supplementary Fig. 11a-c, Effect of Alisertib and Aurora A Inhibitor I on spindle morphology and polarity.

Revised supplementary Fig. 13a-f, Effect of Alisertib and Aurora A Inhibitor I on cell cycle progression and apoptosis.

From our data, both alisertib and Aurora A Inhibitor I showed the phenotypes almost identical to that seen with ENMD-2076 or AURKA siRNA treatment, only except the alisertib's effect on α -tubulin level. Alisertib slightly increased α -tubulin level (Supplementary Fig. 5e, f, 7d), while ENMD2076, Aurora A Inhibitor I and AURKA siRNA all reduced α -tubulin level. However, like ENMD-2076, Aurora A Inhibitor I or AURKA siRNA, alisertib significantly inhibited the microtubule polymerization (Supplementary Fig. 9c and d). These results suggested that, like all other AURKA inhibitors, alisertib inhibition of AURKA led to the destabilization of microtubule, thereby inducing synthetic lethality in RB1 deficient cells. The increase in α -tubulin

level by alisertib was presumably due to its unique off-target effect. Indeed, from a recent DNA-programmed affinity labeling method, α -tubulin was identified as a new alisertib binding protein, in addition to its primary target AURKA (Wang et al, Chem. Eur. J. 2017, 23:10906–10914). A small molecule binding to a protein often causes the stabilization of the binding protein. Hence, we believe that the increase in the level of α -tubulin protein by alisertib was due to the direct binding of alisertib to α -tubulin, which was likely its off-target effect. This is now described in the Results section (pages 5, 7 and 8, marked in red) in the revised manuscript. All the new data were added and figures/texts were rearranged accordingly in the revised manuscript.

In conclusion, I do have significant concerns that need to be addressed before recommending it for publication in Nat Communications.

We are again grateful to the reviewer for all the constructive comments to improve the quality of our study. We hope that the revised version of the manuscript has been significantly improved to meet the standard of the reviewer and the journal. If there is any additional questions or concerns from our revision, we will be happy to address.

REVIEWERS' COMMENTS:

Reviewer #1 (Remarks to the Author):

The authors have addressed my comments more than adequately.

Reviewer #2 (Remarks to the Author):

The authors have addressed all the main concerns and I am supportive of publishing the revised manuscript at Nature Communications.